# Geometric transformation and three-dimensional hopping of Hopf solitons

Jung-Shen B. Tai[1], Jin-Sheng Wu [1] & Ivan I. Smalyukh [1,2,3 ✉]

Arising in many branches of physics, Hopf solitons are three-dimensional particle-like field distortions with nontrivial topology described by the Hopf map. Despite their recent discovery in colloids and liquid crystals, the requirement of applied fields or confinement for stability impedes their utility in technological applications. Here we demonstrate stable Hopf solitons in a liquid crystal material without these requirements as a result of enhanced stability by tuning anisotropy of parameters that describe energetic costs of different gradient components in the molecular alignment field. Nevertheless, electric fields allow for inter-transformation of Hopf solitons between different geometric embodiments, as well as for their three-dimensional hopping-like dynamics in response to electric pulses. Numerical modelling reproduces both the equilibrium structure and topology-preserving out-of-equilibrium evolution of the soliton during switching and motions. Our findings may enable myriads of solitonic condensed matter phases and active matter systems, as well as their technological applications.

[1] Department of Physics and Chemical Physics Program, University of Colorado, Boulder, CO 80309, USA. [2] Department of Electrical, Computer, and Energy Engineering, Materials Science and Engineering Program and Soft Materials Research Center, University of Colorado, Boulder, CO 80309, USA. [3] Renewable and Sustainable Energy Institute, National Renewable Energy Laboratory and University of Colorado, Boulder, CO 80309, USA. ✉email: ivan.smalyukh@colorado.edu

Three-dimensional (3D) topological solitons are marvels of mathematical physics that arise in theoretical models in diverse physical systems including elementary particle and nuclear physics, condensed matter, and cosmology[1–5]. A particularly interesting type of them, the Hopf soliton, is described by the mathematical Hopf map from a hypersphere to an ordinary sphere[6] ($\mathbb{S}^3 \to \mathbb{S}^2$), which in the physical 3D space exhibits interlinked circle-like or knotted localized regions of constant order parameter values mapping to a single point in the order-parameter space[2–4]. In condensed matter systems, materials with order parameters being unit-vector fields (order-parameter space $\mathbb{S}^2$), such as magnetization and electric polarization, or head-tail symmetric director fields (order-parameter space $\mathbb{S}^2/\mathbb{Z}_2$), such as molecular alignment in liquid crystals (LCs), are candidates for hosting Hopf solitons. While the nontrivial nature of $\pi_3(\mathbb{S}^2) = \mathbb{Z}$ and $\pi_3(\mathbb{S}^2/\mathbb{Z}_2) = \mathbb{Z}$ groups informs about the existence of the corresponding topologically nontrivial constructs, it does not guarantee their stability. In fact, the Derrick-Hobart theorem predicts that such solitons cannot be stabilized within the simplest linear field theories[7,8]. Nevertheless, by invoking chirality and a corresponding free energy term as the mechanism of overcoming the constraints of the Derrick-Hobart theorem in LCs and magnets[9], recent discoveries of stable Hopf solitons include hopfions hosted in a uniform background (constant order parameter far-field $n(r) \equiv n_0$; Fig. 1a, e, h, k) and the so-called heliknotons in a helical background ($n(r)$ perpendicular to and twisting around the helical axis $\chi_0$; Fig. 1b, f, i, l), where they were found both individually and within triclinic 3D lattices[10–17]. Hopf solitons were also modeled in conical backgrounds at externally applied fields, where $n(r)$ is at a cone angle $0° < \theta_c < 90°$ with respect to $\chi_0$[17] (Fig. 1c, g, j, m). Localized structures of Hopf fibration[18] and structures resembling heliknotons[19], albeit with singular defects, were also found. In all cases, Hopf solitons classified by the third homotopy group exhibit interlinked regions of constant order parameter (preimages) with conserved linking number, identified with their integral topological charge – Hopf index $Q$ (Fig. 1e, f, g). Another notable feature of topology of Hopf solitons in elastically isotropic materials is that the streamlines of skyrmion number density $\Omega$ (or emergent magnetic field in magents[20,21]) also nontrivially link into Hopf fibrations[13,17], and the surfaces of constant magnitude of skyrmion number density form tori or handlebodies (Fig. 1h–j, Supplementary Movie 1). This is of interest for novel 3D spintronic applications as the emergent fields describe the interaction between magnetic solitons and the spin currents[20,22]. Topological solitons attract fundamental as well as technological interest because of their topology-preserving multi-stability, field driven dynamics, and ability to act as individual particles or even form crystals[16,22–26]. Compared to their lower-dimensional counterparts – 2D skyrmions, Hopf solitons have an advantage that they can be controlled in all three spatial directions.

Beyond chirality, the stability of Hopf solitons in experiments so far has also relied on geometrical confinement or externally applied fields. For example, hopfions in LCs, LC colloids, and magnets have been realized in thin-film and nanodisk geometries where surface boundary conditions (BCs) were shown to be essential[10–15]. Heliknotons in LCs, on the other hand, can be stable in the bulk without confinement, but an external electric field was needed to sustain their stability[16]. While heliknotons in chiral magnets[17] and Hopf solitons as skyrmion knots in frustrated magnets[27] have been shown in numerical simulations to be stable per se, their experimental realizations are lacking. As a result, the 3D mobility and control of Hopf solitons is so far not fully utilized, precluding their technological applications. Moreover, though verified to be topologically identical, hopfions and heliknotons are distinct embodiments of Hopf solitons in their field configurations and are stable under different conditions. Insights into how their field geometries are related to each other are essential for understanding stability of Hopf solitons in and out of equilibrium. In conventional nematic LCs made of rod-like molecules, the twist deformation is favored over bend deformation[28], energetically hindering a smooth transition between the uniform state and the helical state through conical states, as well as the hopfion-heliknoton inter-transformation. However, recently, LCs made of bent-core molecules were found to have exceptionally low bend elastic constant within the nematic phase they form, and mixtures of rod-like and bent-core molecules exhibit tunable elastic anisotropy and are an ideal material system to explore stability of various Hopf soliton embodiments and their geometric inter-transformation[29,30].

In this work, we demonstrate the structural stability of Hopf solitons under different elastic material constants, applied electric field $E$, and confinement conditions in chiral LCs. Our numerical study reveals conditions when Hopf solitons are stabilized at no applied field or confinement in the helical background and when they are stabilized with the help of confinement or applied fields in the uniform or conical backgrounds. Furthermore, we identify a pathway for inter-transformation between hopfions and heliknotons, in confined LCs by switching $E$, where $n(r)$ transforms smoothly while preserving the soliton's topology. We experimentally demonstrate such facile inter-transformation. Our findings indicate that elastic constant anisotropy, tunable by adjusting the material composition, provides a new mechanism for enhancing stability of Hopf solitons in different backgrounds. Further, we have discovered a 3D hopping-like motion of Hopf solitons that arises from repeated inter-transformation between hopfions and heliknotons through periodic voltage switching.

## Results

**Structural stability of Hopf solitons.** Using experimentally determined and then numerically relaxed[11,16] Hopf solitons as initial conditions, we investigated the structural stability of elementary $Q = 1$ Hopf solitons in chiral LCs by minimizing the Frank-Oseen free energy density including the term describing dielectric coupling effect of $E$ (Methods)[16,31,32]

$$f_{\text{CLC}} = f_{\text{elastic}} + f_{\text{electric}} = \frac{K_{11}}{2}(\nabla \cdot n)^2 + \frac{K_{22}}{2}(n \cdot \nabla \times n)^2 \\ + \frac{K_{33}}{2}(n \times \nabla \times n)^2 + \frac{2\pi K_{22}}{p_0}(n \cdot \nabla \times n) - \frac{\varepsilon_0 \varepsilon_a}{2}(E \cdot n)^2 \quad (1)$$

Here $n(r)$ is the LC molecular alignment field; for continuous 3D solitonic excitations, it can be treated as a vector field for simplicity[10,11]. $K_{11}$, $K_{22}$, and $K_{33}$ are Frank elastic constants for splay, twist, and bend deformations, respectively, $p_0$ is the equilibrium pitch of the chiral LC at no field, $\varepsilon_0$ is the vacuum permittivity, $\varepsilon_a$ is the dielectric anisotropy of LC, and $E$ is along $z$. Mimicking our experiments, we consider conditions where solitons can be stabilized in an unconfined bulk (Fig. 2a and Supplementary Fig. 1), for confinement with perpendicular BCs for $n(r)$ along $z$ (Fig. 2b and Supplementary Fig. 2) and for unidirectional parallel BCs (Supplementary Fig. 3) by varying the elastic anisotropy associated with bend and twist deformation ($\sqrt{K_{33}/K_{22}}$) and the normalized electric field strength $\widetilde{E}$ (Methods).

The translationally invariant backgrounds of $n(r)$ that embed Hopf solitons for unconfined chiral LCs can be derived analytically by comparing their corresponding free energies (Methods), as shown with the help of $\theta_c$ contours in Fig. 2a. Here $\theta_c$ is the cone angle between the background $n(r)$ and $+\hat{z}$ direction, which is also the direction of $\chi_0$ and $E$ (Fig. 1c). After

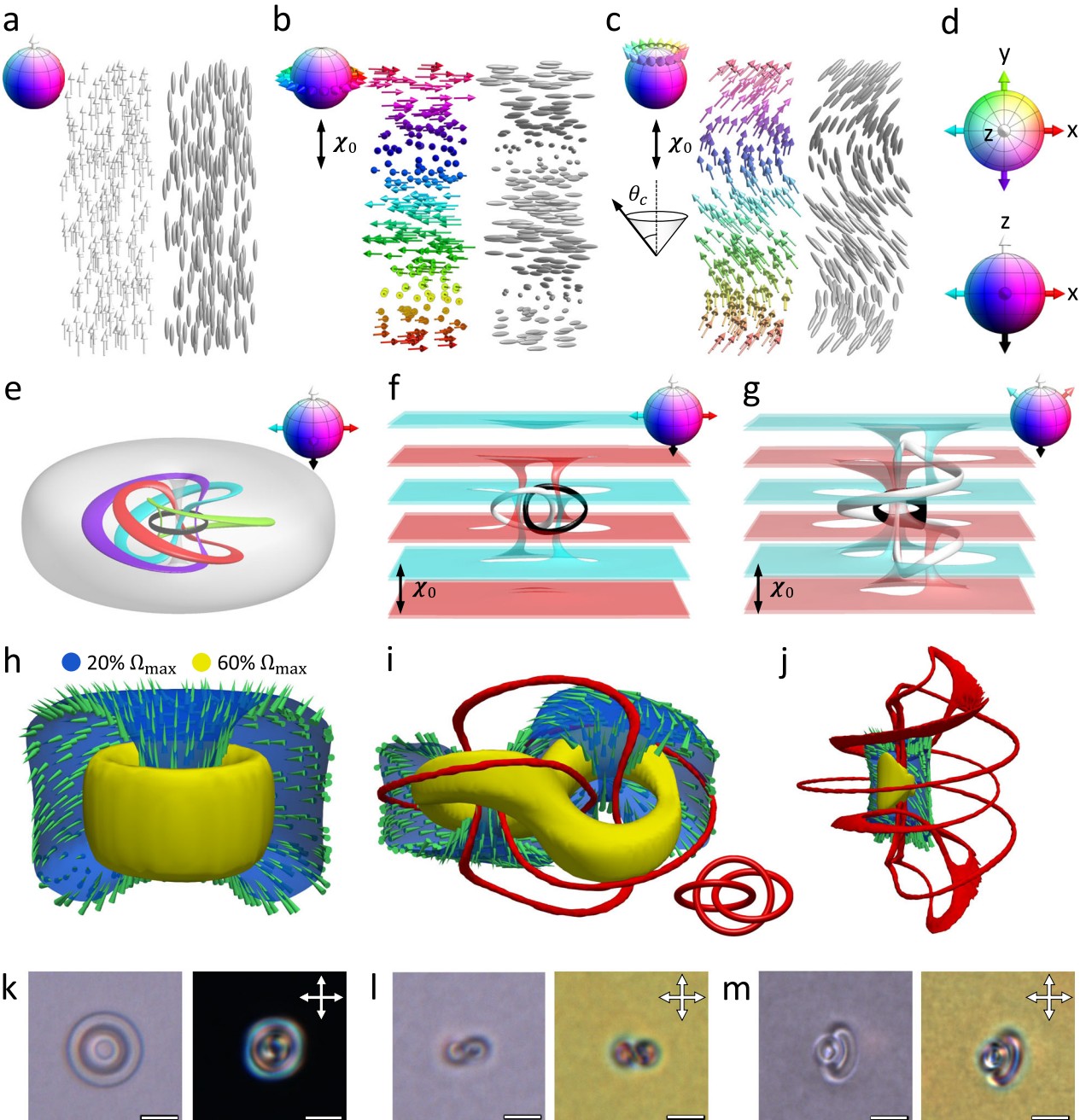

**Fig. 1 3D Hopf solitons in topologically trivial backgrounds. a–d** Schematics of $\boldsymbol{n(r)}$ backgrounds in uniform (**a**), helical (**b**), and conical (**c**) states represented by arrows for vector fields and ellipsoids for nonpolar director fields, respectively. The conical state is at a cone angle $\theta_c$ with respect to the helical axis $\chi_0$. Vectors are colored based on their orientations as shown in the order-parameter sphere in insets and in (**d**). **e–g** Preimages of vector orientations (shown in insets as arrows in the order-parameter sphere) of Hopf solitons stabilized in a uniform (**e**), helical (**f**), and conical (**g**) background. **h–j** Visualizations of the skyrmion number density $\Omega$ and vortex lines of stabilized Hopf solitons in different backgrounds corresponding to **e–g**, respectively. The inset in **i** shows vortex line in the heliknoton can form mutually linked rings[17]. **k–m** Hopf solitons observed in chiral LCs using bright-field (left panels) and POMs (right panels) in a uniform (**k**), helical (**l**), and conical (**m**) background, respectively. The directions of cross polarizers are shown in the insets and scale bars are 5 μm.

energy minimization, Hopf solitons can preserve topology and exist as stable or metastable structures when the energy of the stabilized soliton is lower or higher than the embedding background, respectively; metastable Hopf solitons appear as particle-like, spatially-localized structures that are robust under the effects of thermal fluctuations due to the energetic barrier between these metastable structures and the corresponding stable states (Fig. 2). Parameter regions where the final energy-minimizing structure is topologically trivial or contains singular defects are classified as unstable. At $\widetilde{E} = 0$, Hopf solitons are metastable in the helical background (heliknotons) with higher energy density than the background when $\sqrt{K_{33}/K_{22}} \leq 1.2$ and unstable otherwise (Fig. 2a, e). At larger $\sqrt{K_{33}/K_{22}}$, heliknotons can remain metastable if $\widetilde{E}$ is applied, consistent with previous findings that heliknotons required a stabilizing $\widetilde{E}$ in chiral LCs

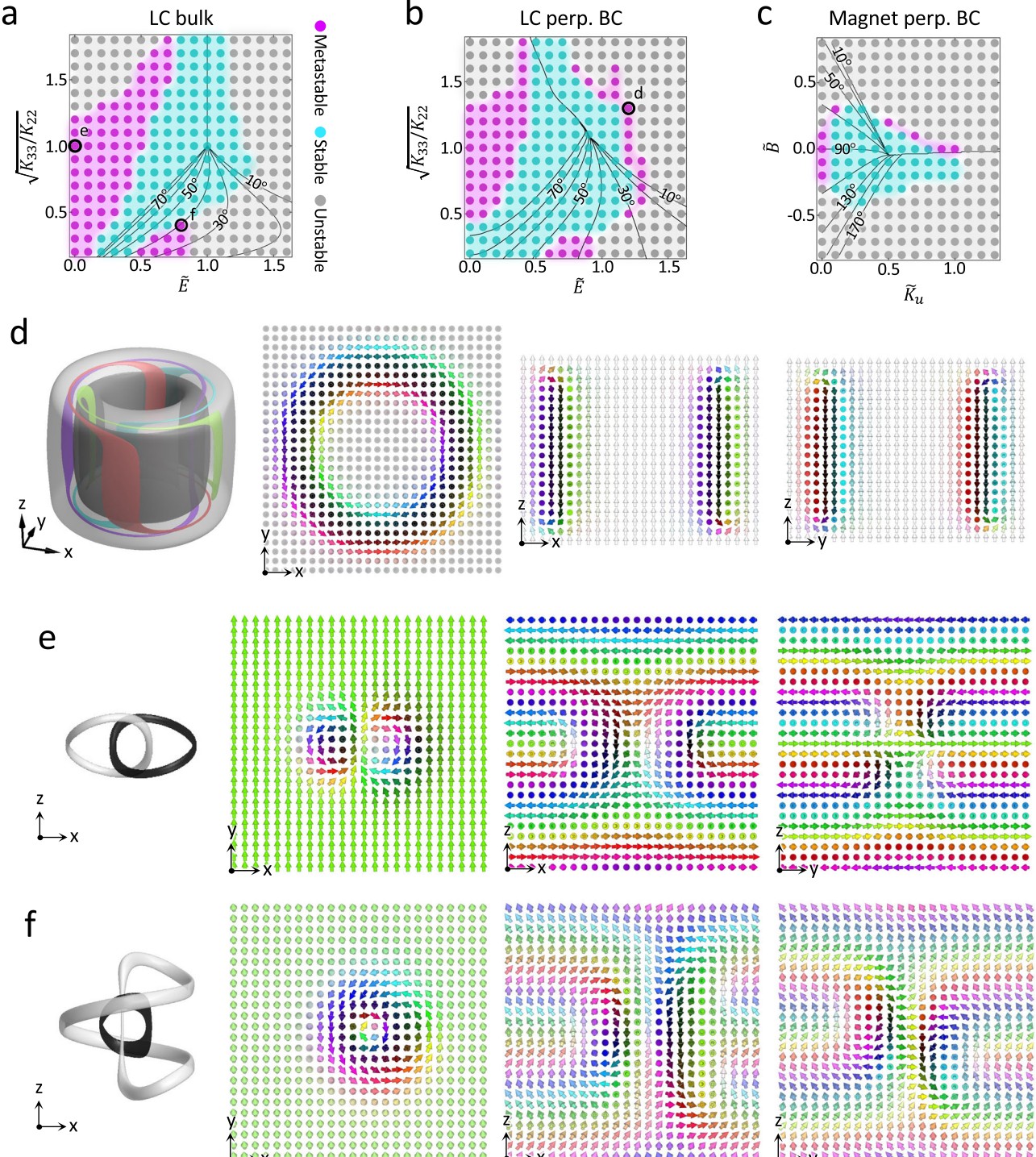

**Fig. 2 Structural stability of Hopf solitons. a–b** Structural stability diagrams of LC Hopf solitons in the bulk (**a**) and within a confined volume with perpendicular BC and $d = 3p_0$ (**b**). The data points are colored based on the stability of Hopf solitons, and the contour lines of background mid-plane cone angle $\theta_c$ are shown on each diagram. $\textbf{n}(\textbf{r})$ of labeled data points in (**a–b**) are shown in (**d–f**). **c** Structural stability diagram of magnetic Hopf solitons in a confined volume with perpendicular BC and $d = 3\lambda$. **d–f**, Preimages and director fields of representative Hopf solitons in a uniform (**d**), helical (**e**), and conical background (**f**), respectively. Preimages and vectors are colored based on the color scheme shown in Fig. 1d.

with $\sqrt{K_{33}/K_{22}} > 1.5$[16]. Hopf solitons can also be metastable in the bulk conical background for $30° \lesssim \theta_c \lesssim 40°$ at $\sqrt{K_{33}/K_{22}} \leq 0.4$ and $0.5 \leq \tilde{E} \leq 0.8$ (Fig. 2a, f). Excitingly, Hopf solitons are found to have $\textbf{n}(\textbf{r})$ with lower energy than their embedding backgrounds between patches of parameter regions of metastable Hopf solitons and across the helical-uniform and helical-conical boundaries for

the background field in the structural stability diagram. In these parts of the diagram, the Hopf solitons fill the computational volume and form stable crystalline assemblies (Fig. 2a). The lower-than-background energy of Hopf solitons results from the unconstrained structural degrees of freedom in the solitonic structures, which become lower in energy density in the intermediate parameter regions than the background states

whose structural degrees of freedom is limited ($\theta_c$ and equilibrium pitch $p$ as functions of $\widetilde{E}$ and $\sqrt{K_{33}/K_{22}}$). Solitons with lower energy density thus form crystalline assemblies to lower the overall free energy. Similar stable assemblies of Hopf solitons have been found in LCs and magnets, albeit here they can occur even without confinement or applied fields[16,17]. Importantly, regardless of the distinct embedding backgrounds with different $\theta_c$, the Hopf indices of the stabilized solitons remain unchanged, as evident from the linking number of preimages (Fig. 2d–f).

When confinement and perpendicular BC were applied to the chiral LC with a thickness $d = 3p_0$, the background $\theta_c$ contours shifted (Fig. 2b). Here, due to confinement, $\theta_c$ is defined as the cone angle the background $\boldsymbol{n(r)}$ make with $+\hat{z}$ direction in the $xy$ midplane of the volume. The parameter regions of stability and metastability of Hopf solitons found in the bulk LC diagram are present also in the case of confinement. However, additional metastability regions of Hopf solitons emerge in the uniform background with $\theta_c = 0°$ (hopfions, Fig. 2d) under the confinement, differing from the case of no hopfions in the fully unwound LC bulk shown in Fig. 2a. This further demonstrates how confinement of LCs at surfaces with perpendicular BCs helps stabilize hopfions[10,11,13]. Remarkably, our structural stability diagram reveals that, with confinement and perpendicular BCs, hopfions and heliknotons can be metastable at different $\widetilde{E}$ at $0.8 \lesssim \sqrt{K_{33}/K_{22}} \lesssim 1.3$, suggesting an in situ pathway for intertransformation by varying the applied voltage. Beyond their structural stability region, Hopf solitons either collapse into the topologically trivial background through proliferation and annihilation of singular defects[12] or transform into point-defect-dressed solitonic structures such as torons[33] (Supplementary Figs. 1 and 2). Hopf solitons in confinement with parallel BCs show structural stability similar to that in bulk LCs (Supplementary Fig. 3).

To understand if similar structural stability and structural transformations of Hopf solitons can also occur in other material systems, we performed stability analysis for Hopf solitons in the magnetization field $\boldsymbol{m(r)}$ of chiral magnets with perpendicular BCs at confining surfaces (Fig. 2c and Supplementary Fig. 4). The micromagnetic Hamiltonian of chiral magnets resembles the Frank-Oseen elastic free energy in the isotropic elasticity limit of $K_{11} = K_{22} = K_{33} \equiv K$ in Eq. (1), with $K$ and $2\pi K/p_0$ related to the exchange and Dzyaloshinskii-Moriya interaction constants, respectively (Methods), suggesting similar energetics and structures can be anticipated in these distinct physical systems[10,23]. We subject this chiral magnetic material to an applied magnetic field $\boldsymbol{H}$ (normalized field strength $\widetilde{B}$) and a uniaxial magneto-crystalline anisotropy (normalized anisotropy strength $\widetilde{K_u}$), both along $\hat{z}$, which couple to $\boldsymbol{m(r)}$ linearly and quadratically (Methods). We find that magnetic heliknotons can be metastable at zero $\widetilde{K_u}$ and $\widetilde{B}$, while finite $\widetilde{K_u}$ promotes a uniform background and metastable magnetic hopfions under confinement (Fig. 2c), similar to the structural stability of Hopf solitons in LCs at $\sqrt{K_{33}/K_{22}} \approx 1$ (Fig. 2b). Through Zeeman coupling, $\boldsymbol{H}$ aligns $\boldsymbol{m(r)}$ linearly, and Hopf solitons can be metastable or stable for either direction of $\boldsymbol{H}$ in a helical or conical background, while hopfions in a uniform background are metastable only when $\boldsymbol{H}$ is in the same direction as the BC magnetization on the surfaces (along $+\hat{z}$). Outside of the stability regions of Hopf solitons, magnetic torons form with $\boldsymbol{m(r)}$ resembling the $\boldsymbol{n(r)}$ of LC torons. As with previously studied magnetic Hopf solitons, the streamlines of skyrmion number density $\boldsymbol{\Omega}$ (or emergent magnetic field) derived from equilibrated Hopf solitons form Hopf fibration, regardless of the embedding background or confinement conditions[13,17] (Fig. 3).

**Geometric transformation of Hopf solitons driven by electric field.** Inspired by the numerically revealed pathway for inter-transforming LC heliknoton and hopfion, we performed simulations and experiments in a confined chiral LC to demonstrate this. Simulation results show that under confinement with perpendicular BCs for $\sqrt{K_{33}/K_{22}} \lesssim 1.1$, the helical state in $\boldsymbol{n(r)}$ can transition smoothly, through conical state, to the uniform state by increasing $\widetilde{E}$ (Fig. 4a). Concomitantly, a heliknoton transforms smoothly into a hopfion (Fig. 4b–d and Supplementary Fig. 5). We experimentally generated a heliknoton with laser tweezers in a confined ($d = 3p_0$) chiral LC mixture with reduced $\sqrt{K_{33}/K_{22}}$ (Fig. 4e, Supplementary Fig. 6, Methods), which remained stable at no $\widetilde{E}$. Upon increasing the voltage applied across the LC cell, the heliknoton-to-hopfion transformation was revealed by bright-field and polarizing optical microscopies (POMs) (Fig. 4e and Supplementary Fig. 7). Notably, though hopfions and heliknotons have been known to share the same $\boldsymbol{n(r)}$ topology[16], this is the first time the two are shown as geometric embodiments of the same field topology through direct imaging of their inter-transformations. This process can be paralleled with transformations between geometric shapes of objects that preserve topological invariants, like the transformation between a coffee mug and doughnut that retains the genus-one surface topology. Remarkably, this transformation is reversible when the voltage is switched off and a heliknoton can undergo a full cycle of transformation to a hopfion and back to heliknoton (Fig. 4e, Supplementary Fig. 7, Supplementary Movies 2-4). The full-cycle transformation is also accurately captured by simulations (Fig. 4f–g, Supplementary Fig. 5, Supplementary Movie 5). A close inspection of the POM micrographs and simulated preimages of the Hopf soliton during transformation show that the full-cycle transformation process of $\boldsymbol{n(r)}$ is nonreciprocal (Figs. 4e–g and 5).

The elapsed time needed to transform from one geometric embodiment (e.g., hopfion or heliknoton) to the other is also asymmetric, with the transformation from a hopfion to a heliknoton roughly three times longer than in the opposite direction (Fig. 4e). The full-cycle transformation is accompanied by a displacement of $\sim 1.2$ µm along the long axis of the heliknoton, or $\sim 0.52$ of the helical pitch $p_0$ of the chiral LC, which is also quantitatively reproduced in simulations (Fig. 4f-g). Throughout the transformation, the field topology of the Hopf soliton is preserved, evident by the linking number of preimages (Figs. 4f and 5, and Supplementary Movie 5). Additionally, we identify Hopf solitons embedded in a conical background at an intermediate $\widetilde{E}$ with an intermediate state of the transformation, showing how the inter-transformation between hopfion and heliknoton progressed through an intermediate stage of a Hopf soliton in the conical background (Supplementary Fig. 8).

**3D hopping and squirming of Hopf solitons.** The heliknoton-hopfion intertransformation and the ensuing spatial displacement repeat with periodic switching of the applied voltage on and off, leading to an activated propelling motion (Fig. 6a, b, Supplementary Movie 6), similar to the squirming motion of 2D skyrmions[34]. This translational motion is enabled by the non-reciprocal evolution of $\boldsymbol{n(r)}$ during the inter-transformations between the various geometric embodiments of the Hopf soliton with the voltage modulation within each modulation period. Both experiments and simulations show the consistent direction of displacement, suggesting the long axis of the heliknoton embodiment of the inter-transforming Hopf soliton is oriented. This is because the presence of confinement and perpendicular BCs preselect one of the two polar preimages ($\pm\hat{z}$; white and back in

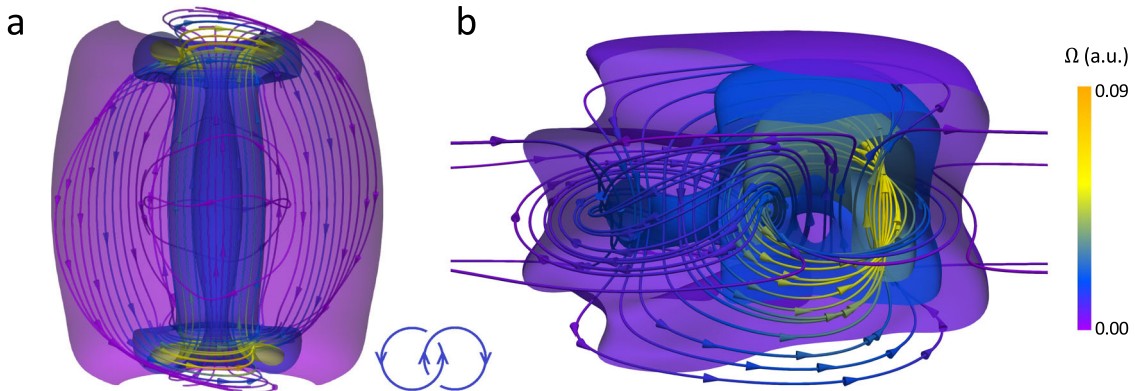

**Fig. 3 Interlinked streamlines of skyrmion number density in magnetic Hopf solitons.** Skyrmion number density $\Omega$ shown by streamlines and magnitude isosurfaces in Hopf solitons stabilized at $\widetilde{B} = 0$, $\widetilde{K}_u = 1$ (**a**) and $\widetilde{B} = 0.3$, $\widetilde{K}_u = 0.1$ (**b**), respectively. Cones on the streamlines indicate the local orientation of $\Omega$. The schematic in **a** shows that each pair of closed streamlines has a linking number $+1$.

**Fig. 4 Geometric transformation of Hopf solitons driven by electric field. a** Schematic illustration of the experimental setup and the switching of topologically trivial backgrounds; $\theta_c$ at midplane decreases from 90° to 0° as the applied voltage increases. **b**–**d** Vertical midplane cross-sections through the Hopf soliton going through intertransformation from a heliknoton to a hopfion. **b**–**d** correspond to simulated $\boldsymbol{n}(\boldsymbol{r})$ at 0.00%, 0.19%, 6.07% of the total simulation time in **f**–**g**. The 100% simulation time corresponds to the total inter-transformation time (48.06 s) in experiments. **e** Experimental POM snapshots of Hopf soliton switching from a heliknoton (0 s) at $U = 0$ V to a hopfion (11.73 s) at $U = 3.85$ V, and back to a heliknoton (48.06 s) at $U = 0$ V. $d/p_0 = 3$, $d = 7$ μm and scale bars are 5 μm. **f**–**g** Simulated transformation shown by preimages of two antiparallel orientations (**f**; white: $+\hat{z}$, black: $-\hat{z}$) and simulated POM micrographs (**g**). The vertical red line going through the center of the volume in **f** is a guide to the eye. The progress of inter-transformation in numerical simulations shown as percentage of the total simulation time is labeled in each panel. In simulations, $K_{33}/K_{22} = 1$ and $d/p_0 = 3$.

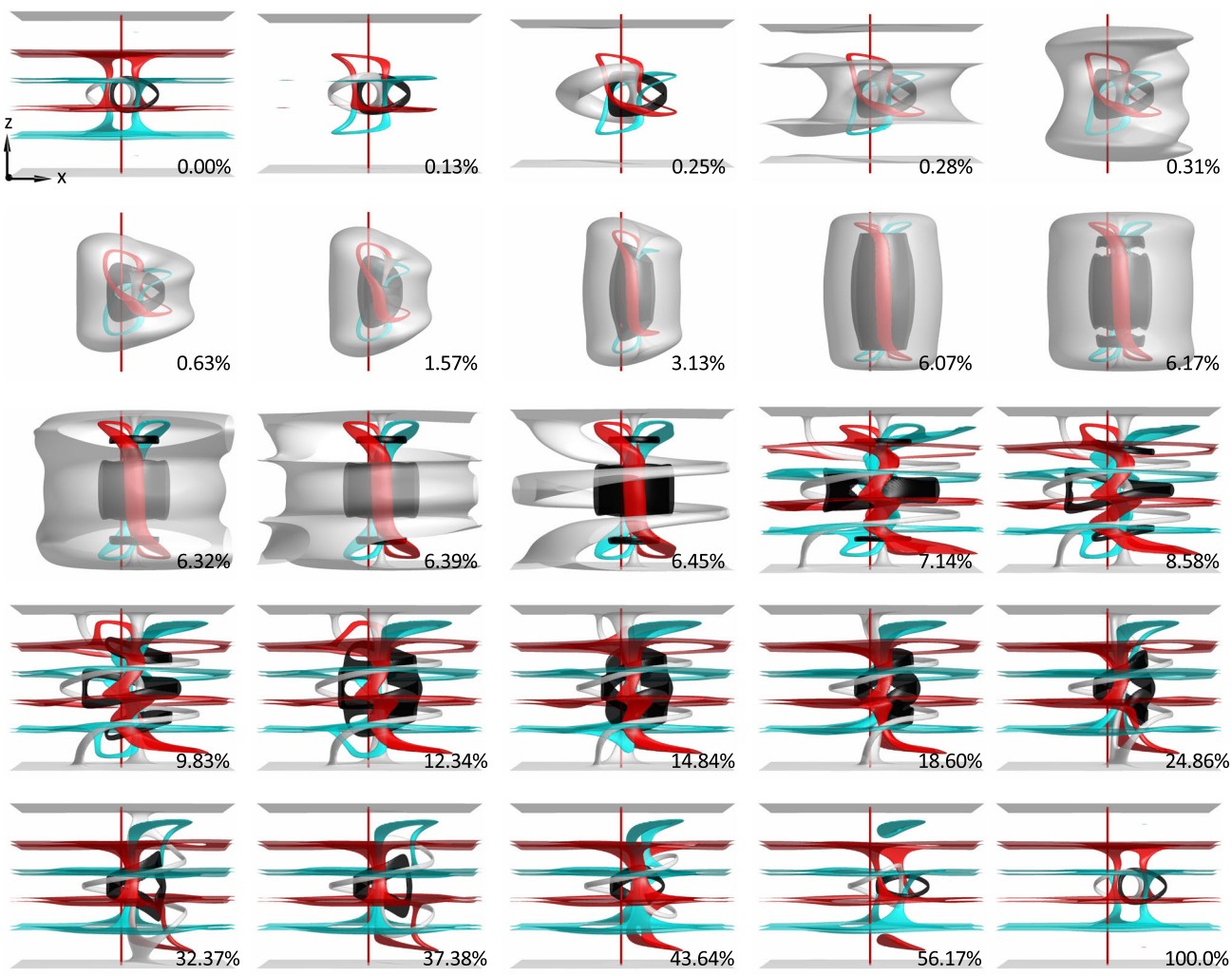

**Fig. 5 Preimage visualization of the simulated geometric transformation of a Hopf soliton.** Simulated geometric inter-transformation of a Hopf soliton from the heliknoton embodiment in the helical background to the hopfion embodiment in the uniform background, and back to the heliknoton embodiment visualized by preimages (white: $+\hat{z}$, black: $-\hat{z}$, red: $+\hat{x}$, cyan: $-\hat{x}$). The corresponding percentage of the total inter-transformation time in simulations is labeled in each panel. The conserved linking number of each pair of preimages demonstrates the conserved field topology of the Hopf soliton during the transformation process.

Fig. 4f) as associated with the far-field upon transitioning into a uniform background by an electric field $\boldsymbol{E}$, thus distinguishing the two polar preimages and orienting the long axis of the heliknoton in a confined LC. We also note that the directions of motion are the same during the heliknoton-to-hopfion and hopfion-to-heliknoton transformations (Fig. 4e–g); this is different from the case of 2D skyrmions that move in opposite directions during the forward and backward transformations under a modulated $\boldsymbol{E}$ but with different magnitudes, giving rise to a rectified motion[34,35]. Experimental and computer-simulated preimage visualizations and POM micrographs, which exhibit a good agreement (Fig. 4e–g), vividly reveal these origins of the nonreciprocal structural evolution and soliton motion.

Interestingly, such translational motion of a Hopf soliton deviates from a linear trajectory. Within each transformation cycle, the Hopf soliton undergoes a net displacement along the direction of heliknoton's long-axis orientation, while the long-axis orientation of the heliknoton with respect to $x$-axis ($\phi$) slowly changes with time (Fig. 6a–c, Supplementary Fig. 9). Moreover, we found heliknotons adopt various orientations even at static equilibrium with no applied voltage and when the substrates are rubbed to achieve a uniform in-plane orientation in the

background helical state at the sample mid-plane (Fig. 6d). Since the orientation of the soliton correlates with its $z$ position along $\boldsymbol{\chi}_0$ in a helical background, as previously revealed by direct 3D imaging[16], this suggests that, beyond 2D lateral dynamics, Hopf solitons also undergo orientation-correlated displacements in the third dimension along $\boldsymbol{\chi}_0$. To further understand the propensity of Hopf solitons to adopt certain orientations and $z$ positions, we numerically investigated the initial ($z_i$) and final, energy-minimizing $z$ positions ($z_f$) with respect to the sample midplane before and after equilibration, as well as the final free energy (Fig. 6e–g). This was done by numerically displacing a hopfion or heliknoton from the sample midplane in their respective backgrounds before relaxing them towards equilibrium (Heliknotons so displaced were also rotated in a way consistent with the helical background; see Methods). Our results revealed that hopfions quickly relaxed and symmetrically filled up the entire vertical space between the confining substrates ($z_f = 0$), regardless of its initial position $z_i$; this yields a linear relation between $z_f - z_i$ and $z_i$ (Fig. 6e). On the other hand, we found the energy landscape of a heliknoton in a helical background with confinement and perpendicular BC contains multiple energy minima for spatial

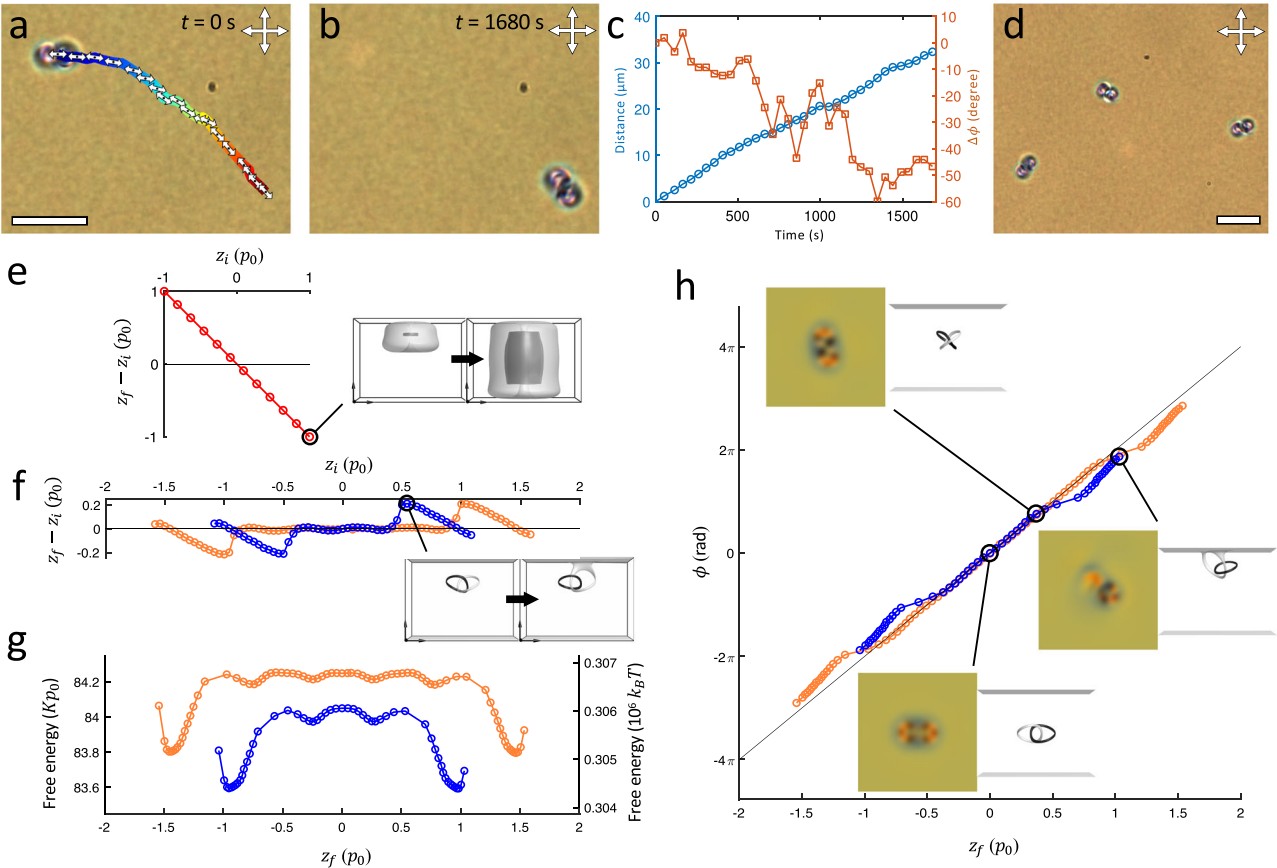

**Fig. 6 3D Hopping of Hopf solitons. a–b** Translational and orientational displacement of a Hopf soliton by repeated voltage switching shown at its initial (**a**) and final (**b**) position. The 2D trajectory is color-coded by time and the long-axis orientations of the Hopf soliton in helical background at intermediate positions are shown by double arrows (plotted for every two switching cycles). **c** Distance and accumulated change in orientation in each transformation cycle ($\Delta\phi$) of a hopping Hopf soliton shown in **a–b**. **d** Hopf solitons in different long-axis orientations in a helical background with perpendicular confinement. **e** Displacement in $z$ as a function of initial position $z_i$ of a hopfion in the uniform background. **f** Displacement in $z$ as a function of initial position $z_i$ of a heliknoton in the helical background. **g–h** Free energy (**g**) and orientation (**h**) dependence of a heliknoton on equilibrium position $z_f$. In **f–h**, heliknotons with $d = 3p_0$ and $d = 4p_0$ are shown in blue and orange, respectively. The insets in **e**, **f** show the initial and final equilibrated Hopf solitons of the corresponding data points by polar preimages. Insets in **h** show the simulated POM images (viewed along $\hat{z}$) and the preimages (viewed along $\hat{y}$) of the solitons with $z$ positions 0, $0.42p_0$, and $1p_0$ relative to the midplane (left to right) for $d = 3p_0$. The line in **h** shows $\phi = 2\pi(z_f/p_0)$ for bulk heliknotons is a guide to the eye. Free energy is in units of $Kp_0$ and $k_BT$, where $K = 6.47$ pN is the average elastic constant of 5CB and $p_0 = 2.33$ μm, $k_B$ is the Boltzmann constant, and temperature $T = 300$ K. $d = 3p_0$ in experiments in **a–d** and scale bars are 10 μm.

locations along $z$. The lowest energy minima were at $\sim 0.5p_0$ away from the confining substrates, whereas local energy minima were distributed symmetrically around the cell midplane, leading to a nontrivial relation between $z_i$ and $z_f$ of the soliton (Fig. 6f, g). The midplane is an unstable position and corresponds to a local energy maximum. As a result, the dependence of heliknoton orientation $\phi$ on $z$ is slightly perturbed from the anticipated linear relation, though it is still monotonic and allows for the $z$ position of a heliknoton to be inferred by its orientation in POM micrographs within the same $2\pi$ rotation period (Fig. 6h). The displacement in $z$ and energy landscape are qualitatively similar for $d = 3p_0$ and $d = 4p_0$, showing these features are general for heliknotons in a helical background with confinement.

Our analysis suggests the following scenario of periodically repeated geometric inter-transformation and activated motion. In each geometrically distinct state as a hopfion, the soliton equilibrates as it moves towards the $z = 0$ midplane in a uniform background when the voltage is on. In the subsequent heliknoton state with the voltage turned off and the background being helical, thermal fluctuations and/or sample imperfections tip it away from the then unstable $z = 0$ position towards the nearest energy minimum accompanied by a rotation. The soliton again

equilibrates towards $z = 0$ in the following hopfion state, and so on and so forth. In this process, the nonreciprocal director evolution causes squirming and the Hopf soliton propels along the lateral directions while, at the same time, moves vertically between energy minima along $z$ – an effective hopping dynamics in the 3D space.

The squirming motion of Hopf solitons does not require complete inter-transformation between hopfion and heliknoton and their respective uniform and helical backgrounds (Fig. 7). In thinner confined chiral LC slabs ($d = 1.7p_0$) with perpendicular BCs, Hopf solitons also propel in response to low-magnitude voltage modulation with extremes corresponding to a hopfion in a uniform background and a perturbed hopfion in a helical background (Fig. 7a–c). We found that such activated motion is characteristic for hopfions with both +1 and -1 Hopf indices (Fig. 7b–e, Supplementary Movies 7 and 8). Numerical modeling shows the details of nonreciprocity in $n(r)$ of temporal evolution of the Hopf soliton during a single period of voltage modulation that drives the activated motion (Fig. 7f).

## Discussion
To conclude, we have shown, for the first time, that Hopf solitons can exist as spatially localized structures without external fields or

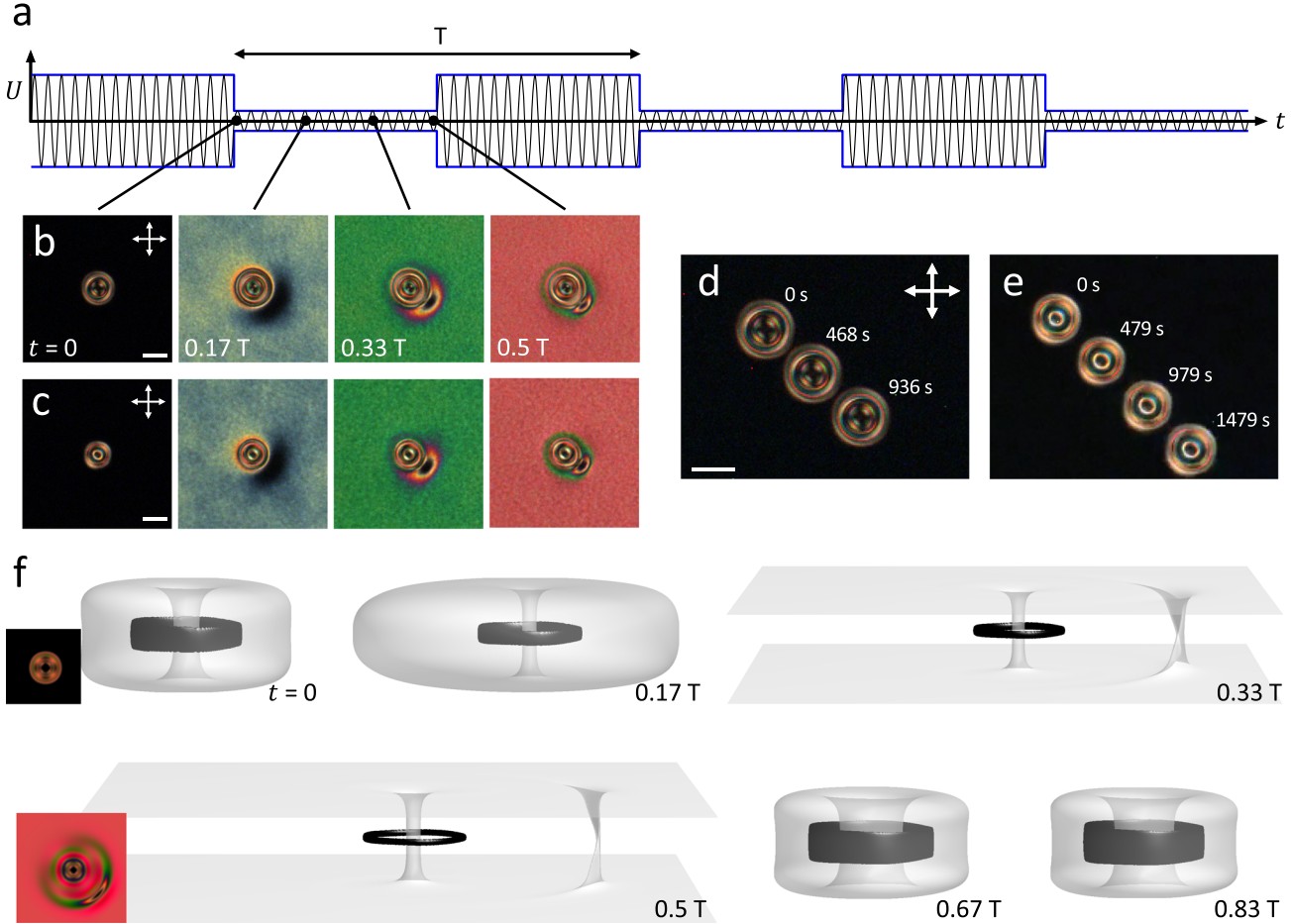

**Fig. 7 Squirming of Hopf solitons. a–c** Snapshots of POMs of a $Q = +1$ (**b**) and a $Q = -1$ (**c**) Hopf soliton subject to background modulation by a modulating voltage profile shown in (**a**). Time points in the modulation period corresponding to snapshots in **b–c** are labeled in **a**. The modulation period $T = 2$ s. **d–e** Squirming of Hopf solitons of Hopf indices $Q = +1$ (**d**) and $Q = -1$ (**e**) shown by superimposed POMs of Hopf solitons at different times. **f** Field configurations of a $Q = +1$ Hopf soliton under voltage modulation in numerical simulations visualized by $\pm\hat{z}$ preimages (in white and black). Shown as insets are the corresponding simulated POM micrographs. $d = 1.7p_0$ and scale bars are 20 μm.

confinement in the helical background of chiral LCs within the proper range of elastic anisotropy parameters, as well as can be stable or metastable in the conical backgrounds when an external electric field is applied. The composition-based engineering of elastic constant anisotropy – energetic costs between different components of the gradient of field – provides a novel route for enhancing soliton stability, beyond the known methods of overcoming constraints of the Derrick-Hobart theorem[3,9,27]. Materials such as novel LCs with exceptionally large elastic-constant anisotropy and their mixtures[29,36,37], as well as magnetic LC colloidal materials based on them[10], can be used as testbeds for the stability of 3D topological solitons, expanding the selection of material systems hosting them. Moreover, we show that, by reducing the bend elastic constant, soliton stability in LCs resembles that in elastically isotropic magnets. This demonstrates elastic anisotropy engineering can further establish LCs as a testbed for general magnetic structures at a quantitative level. We note that the magnitude of splay elastic constant, which represents the other degree of freedom in engineering the elastic-constant anisotropy in LCs, potentially can be additionally adjusted to modulate the stability of Hopf solitons; the independent and coordinated effect of elastic anisotropy engineering on the stability of solitons hosted in these materials warrants further future studies.

We have also unambiguously demonstrated that hopfions and heliknotons are geometric embodiments of the same underlying field topology and can be transformed reversibly between one another by

an electric field, in a way resembling inter-transformations between geometrically different but topologically identical surfaces where the topological invariant, genus, is preserved. Furthermore, using a simple modulating electric field, we demonstrated 3D hopping-like dynamics of Hopf solitons as a result of combined nonreciprocity in the transformation of field configurations and multi-minima energetic landscape in a confined chiral LC. In principle, a more complex modulation profile including asymmetric ON/OFF periods, delays, or arbitrarily shaped profile, may be optimized to control the speed or even the direction of the hopping of Hopf solitons, enabling novel active matter systems made of topological solitons[38]. The newly discovered stability of Hopf solitons and their 3D hopping dynamics is of great interest for technological applications and adds to the diversity of spatio-temporal manipulation methods of topological quasi-particles.

## Methods
**Preparation of samples**. Homogeneous mixtures of 4-Cyano-4'-pentylbiphenyl (5CB, from EM Chemicals) and 4',4"-(heptane-1,7-diyl)-dibiphenyl-4-carbonitrile (CB7CB; from SYNTHON Chemicals, Germany) were obtained by mixing the two compounds at 125°C in the isotropic phase with active stirring. The resulting mixtures at 60 or 70 wt% of 5CB were in nematic phase at room temperature[30]. The nematic mixtures were then added with a small amount of right-handed chiral additive CB-15 (EM Chemicals) to achieve right-handed chiral LC mixtures with helical pitch $p_0$ ranging from 2.33 to 10 μm as measured in a Grandjean-Cano wedge cell[39].

The LC cells were made with Indium Tin Oxide (ITO) coated glass slides or coverslips. The ITO glasses were treated with polyimide SE5661 (Nissan Chemicals) by spin-coating at 2,700 rpm for 30 s and then baking at 90°C, followed by 1 h at 180°C to set strong perpendicular boundary conditions for $n(r)$ at the LC-glass interface. The polyimide-treated ITO glasses were then rubbed mildly and assembled in antiparallel rubbing directions to ensure the background $n(r)$ is a uniform helical state with constant in-plane orientation at sample midplane at no electric field (Fig. 4a). To assemble into a cell, silica micro-cylinders with diameters 7 μm to 20 μm were used as spacers and were sandwiched between ITO glasses and fixed by UV-curable glue. Metal wires were additionally soldered to ITO glasses as electrodes for electric control. Chiral LC mixtures were then introduced into the cell by capillary forces.

To achieve electric control of LC background fields and solitons, the electrodes of the LC cells were connected to a function generator (DS345; Stanford Research Systems) operating at 1 kHz carrier frequency with sinusoidal output to preclude complex hydrodynamic effects[31]. Additionally, we used an in-house MATLAB code controlling a data acquisition board (NIDAQ-6363, National Instruments) connected to a computer for fast modulation of voltage output.

**Laser generation and imaging of Hopf solitons.** Hopf solitons were generated by holographic laser traps capable of producing predesigned patterns of laser intensity within the LC sample[16]. The tweezers setup is based on an ytterbium-doped fiber laser (YLR-10-1064, IPG Photonics, operating at 1,064 nm) and a phase-only spatial light modulator (P512-1064, Boulder Nonlinear Systems) integrated with an inverted optical microscope (IX81, Olympus)[33]. Upon focusing the 1,064 nm laser into the LC sample, local heating and optical realignment created initial $n(r)$ that eventually relaxed into Hopf solitons under suitable energetic conditions[16].

Bright-field microscopy, polarizing optical microscopy and videomicroscopy were performed using the IX-81 Olympus inverted microscope and a charge-coupled device camera (Flea-COL, from PointGrey Research)[33]. Phase-contrast microscopy was performed using a condenser annulus and a 60X oil-immersion phase contrast objective. Differential interference contrast microscopy was performed by introducing Nomarski prisms into the light path between crossed polarizers.

Three-photon excitation fluorescence polarizing microscopy (3PEF-PM) imaging of $n(r)$ in Hopf solitons was performed by a setup built around the same IX-81 microscope integrated with bright-field microscopy and laser tweezers[16,34]. 5CB molecules in the LC mixture were excited via three-photon absorption by using a Ti-Sapphire oscillator (Chameleon Ultra II; Coherent) operating at 900 nm with 140-fs pulses at a repetition rate of 80 MHz[40]. The fluorescence signal was epi-collected by an 100X oil-immersion objective with NA = 1.4 and detected through a 417/60-nm bandpass filter by a photomultiplier tube (H5784-20, Hamamatsu). The polarization state of the excitation beam was controlled by using a polarizer and a rotatable half-wave retardation plate or a quarter-wave retardation plate. In 3PEF-PM imaging experiments, a third-order nonlinear optical process was involved, and the image intensity scales as $\cos^6\beta$, where $\beta$ is the angle between the dipole moment of the LC molecule, orientating along $n(r)$, and the polarization of the exciting light. 3PEF-PM reveals the 3D $n(r)$ field in LCs and were used to unambiguously confirm the geometry and topology of the observed solitonic field configurations[10–12,16].

Computer-simulated polarizing optical micrographs were based on the Jones matrix approach[12,33] for energy-minimizing $n(r)$ configurations and by using the optical birefringence value of 5CB (Δn = 0.18). Computer simulations of the 3PEF-PM images were based on the $\cos^6\beta$ dependence in the image intensity, using the $n(r)$ of Hopf solitons from energy minimizations.

**Numerical modelling.** We used numerical modelling based on energy minimization to explore the stability of Hopf solitons in chiral LCs and chiral magnets. For a chiral LC subjected to an external electric field, the total free energy density consists of Frank-Oseen elastic terms and the dielectric coupling term as shown in Eq. (1). Surface anchoring (anisotropic surface energy term accounting for preferred $n(r)$ orientation at the surface) and saddle-splay deformation terms were not included in our modeling due to strong anchoring strength at the boundaries achieved in experiments. The material parameters were chosen to match those of 5CB except for $K_{33}$, which was a variable parameter to account for the change in bend elastic constant enabled by varying the composition of LC mixture; namely, $K_{11} = 6.4$ pN, $K_{22} = 3$ pN, and $\varepsilon_a = 13.8$[16]. We used a normalized electric field strength $\widetilde{E} = \sqrt{\frac{\varepsilon_0\varepsilon_a}{K_{22}}\left(\frac{p_0}{2\pi}\right)^2} E$ in our modeling and structural stability diagrams such that the first-order transition between the helical and the uniform state happens at $\widetilde{E} = 1$ for bulk LCs when $\sqrt{K_{33}/K_{22}} > 1$ (see below). The energy was iteratively minimized using an energy-minimization routine with finite-difference discretization in space and forward Euler method in time implemented in an in-house MATLAB code[16,32]. Briefly, $n(r)$ was updated iteratively from an initial structure using the Euler-Lagrange equation derived from Eq. (1). Relaxation was terminated when the change in the spatial average of functional derivatives, between iterations, converged and dropped below a threshold value determined for the steady-state stopping criterion, indicating an energy minimum is attained. In all simulations, the computational volume was sampled isotropically by a cubic grid at 24 gird points per $p_0$.

In bulk LC, topologically trivial background fields at different $K_{33}/K_{22}$ and $\widetilde{E}$ can be derived analytically by energy-minimizing the ansatz for a general conical state twisting around $\chi_0$ along $\hat{z}$ with variable pitch $p$ and cone angle $\theta_c$: $n(r) = \cos(2\pi z/p)\sin(\theta_c)\hat{x} + \sin(2\pi z/p)\sin(\theta_c)\hat{y} + \cos(\theta_c)\hat{z}$[41]. For $\sqrt{K_{33}/K_{22}} > 1$, no conical state is stable, and a first-order phase transition boundary exists at $\widetilde{E} = 1$ between the helical and uniform states. Conical states emerge when $K_{33}/K_{22} \leq 1$ and $\widetilde{E} \in [\widetilde{k}, 1/\widetilde{k}]$ and the corresponding equilibrium pitch and cone angle are $p = p_0\widetilde{k}/\widetilde{E}$ and $\cos^2(\theta_c) = \frac{1-\widetilde{k}/\widetilde{E}}{1-\widetilde{k}^2}$, where $\widetilde{k} \equiv \sqrt{K_{33}/K_{22}}$. The structural diagram of trivial background states in bulk LC is shown in Fig. 2a by $\theta_c$ contours. For the background fields of LCs in confinement, 1D simulations along z (translationally invariant in x and y) with Dirichlet boundary conditions were performed. $\theta_c$ was measured at the xy-midplane of the volume to yield the background $\theta_c$ contours shown in Fig. 2b, c and Supplementary Figs. 2-4.

To model solitons in bulk LCs without confinement (Fig. 2a, Supplementary Fig. 1), the size of the computational volume was $4p_0$ in x and y and thickness $d = 10p$ in z. Note that since $p$ depends on $\widetilde{E}$ in conical state, $d$ has to be an integral number of the equilibrium pitch $p$ (a function of $\widetilde{k}$ and $\widetilde{E}$) to avoid artificial frustration caused by the finite computational volume. Analytically derived $\theta_c$ was used as the boundary condition at the top and bottom surfaces. For solitons in a confined volume, Dirichlet boundary conditions of either perpendicular (along $\hat{z}$, Fig. 2b, c, Supplementary Figs. 2 and 4) or unidirectional parallel (along $-\hat{y}$, Supplementary Fig. 3) alignment at top and bottom surfaces were implemented. Periodic boundary conditions were implemented in x and y directions for all simulations. Two initial conditions of $n(r)$ were constructed by either inserting a previously relaxed hopfion[11] into a uniform background with $n_0$ parallel to $\hat{z}$ or a heliknoton[16] into a helical or conical background with $\chi_0$ parallel to $\hat{z}$. The topology of the steady states after energy relaxation was analyzed and a data point is marked with $Q = 1$ if a Hopf soliton was stabilized with one of the initial conditions. Hopf index $Q$ was determined by both the linking number of preimages and numerical integral of the relaxed $n(r)$[12,16]. In all cases the two methods yielded consistent results except when $n(r)$ contained singular defects and a Hopf index cannot be properly defined. The nonpolar chirality axis field $\omega(r)$ was derived from the as relaxed soliton structure by identifying the chirality axis at all spatial coordinates with the eigenvector of the local chirality tensor $C_{ij} = n_k\epsilon_{ljk}\partial_l n_l$[16]. The singular vortex lines in $\omega(r)$ were determined by finding connected spatial regions where $\omega(r)$ is ill-defined.

The algorithm for energy minimizing solitons in chiral magnets is the same as described above for chiral LCs. The micromagnetic Hamiltonian density of a chiral magnet under an external magnetic field and uniaxial magnetocrystalline anisotropy (Fig. 2c, Supplementary Fig. 4) reads[13,17]

$$f_{\text{magnet}} = \frac{J}{2}(\nabla \cdot m)^2 + D(m \cdot \nabla \times m) - \mu_0 M_s m \cdot H - K_u(m \cdot l_0)^2 \qquad (2)$$

with $m(r)$ the magnetization field, $J$ the exchange constant, $D$ the Dzyaloshinskii-Moriya interaction constant, $\mu_0$ the vacuum permeability, $M_s$ the saturation magnetization, $H$ the applied magnetic field, and $K_u$, $l_0$ the strength and direction of bulk uniaxial anisotropy, respectively. $H$ and $l_0$ were both along $\hat{z}$. The similarity between the continuum energy functionals of LCs and magnets suggest that, at some level, similar structures and phenomena can be anticipated in these distinct physical systems, though it should be noted either system has physical properties unique to itself that can contribute additional relevant free-energy terms and distinct phenomena. For example, additional terms in the micromagnetic Hamiltonian can arise from magnetocrystalline anisotropies, Zeeman energy coupling to external fields, and nonlocal dipole-dipole interaction. Here, the nonlocal dipole-dipole interactions were neglected for simplicity, as often done in literature[17,42,43]. The computational volume was parameterized by helical wavelength $\lambda \equiv 2\pi J/D$ (equivalent to $p_0$ in LCs) and Hamiltonian by dimensionless magnetic field $\widetilde{B} \equiv \mu_0 \frac{M_s J}{D^2} H$ and dimensionless anisotropy strength $\widetilde{K}_u \equiv \frac{J}{D^2}K_u$. The skyrmion number density $\Omega$, which is proportional to emergent magnetic field in magnetic solids, was calculated as $\Omega_i = \frac{1}{8\pi}e^{ijk}m(r) \cdot (\partial_j m(r) \times \partial_k m(r))$, where $e^{ijk}$ is the totally antisymmetric tensor.

To understand the energy landscape and the stability of Hopf solitons at different $z$ positions in a confined chiral LC with perpendicular BCs summarized in Fig. 6e–h, Hopf solitons were displaced from the sample midplane to different $z$ positions before relaxing towards equilibrium. For heliknotons, the $n(r)$ of a localized heliknoton was cropped away from the bulk simulation and displaced $z_i$ along z-axis from the midplane while rotated $2\pi(z_i/p_0)$ in the sense of the material chirality as the initial condition and relaxed at $\widetilde{E} = 0$ and $\sqrt{K_{33}/K_{22}} = 1$, the same parameters used for the initial simulation of heliknoton in the bulk. The thicknesses of the simulated cells were $d = 3p_0$ or $4p_0$. After relaxation, the average position of the geometric centers of polar preimages ($n(r) = \pm\hat{z}$) was used as the final position $z_f$ of a heliknoton, and the in-plane orientation of the vector connecting the geometric centers of the polar preimages was used as heliknoton orientation $\phi$ and agrees with the long axis of a heliknoton in POM images. The geometric centers of each polar preimage associated with the heliknoton was determined by finding the center of the minimum bounding sphere[44] of each preimage after excluding the preimages close to the top and bottom surfaces due to

boundary conditions. Hopfions stabilized in a uniform background in a confined LC (perpendicular BCs) with $d = p_0$ were placed in a $d = 3p_0$ cell at different $z_i$ positions and relaxed at $\tilde{E} = 1.1$ and $\sqrt{K_{33}/K_{22}} = 1$. Free energy of the relaxed solitons was calculated by integrating Eq. (1) and the energy of a bulk helical state at $\tilde{E} = 0$ was taken as the reference.

## Data availability

All data generated or analyzed during this study are included in the published article and its Supplementary Information and are available from the corresponding author on a reasonable request. Data generated in this study are provided in the Supplementary Source Data file.

## Code availability

The codes used for the numerical calculations are available upon request.

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

## Acknowledgements

We acknowledge discussions and technical assistance of Andrii Repula, Mariacristina Rumi, Min Shuai, Bohdan Senyuk, and Timothy White. We are grateful to Patrick Davidson for providing CB7CB used at the initial stages of this study. This research was supported by the U.S. National Science Foundation grant DMR-1810513.

## Author contributions

J.-S.B.T. performed experiments. J.-S.B.T. and J.-S.W. performed numerical modelling. I.I.S. directed research and provided funding. J.-S.B.T., J.-S.W. and I.I.S. wrote the manuscript.

## Competing interests

The authors declare no competing interests.
