## [Peer review file · Nature Communications]

REVIEWER COMMENTS

Reviewer #1 (Remarks to the Author):

They have shown, for the first time, that Hopf solitons can exist as spatially localized structures without external fields or confinement in the helical background of chiral LCs within the proper range of elastic anisotropy parameters, as well as that they can be stable or metastable in the conical backgrounds when an external electric field is applied.

However, it is questionable whether making their previously made Hopf soliton meta stable is noble. Nevertheless, a pathway for inter-transformation between hopfions and heliknotons as well as for 3D hopping-like dynamics have been identified in detail and the simulated data is considerable. Accordingly, I think that an impact can be made in expanding the applicability of Hopf solitons and in engineering related to materials.

I recommend the following needs revision.

Figures:

1. Not only do the structural stability diagrams in Figure 2 lack the definition of stable, meta stable, unstable, but they also do not have a clear explanation of the contour lines of background mid-plane cone angle θ_c .

For example, what does it mean to be in a meta stable state? If the Hopion is created in a E-field through inter-transformation, will it be maintained even if the E-field is turned off? The structural stability diagrams in Fig. 2 lack detail in their explanations.

2. Figure 2a, b does not have descriptions regarding e, d, g in the caption, so their relation with Fig. 2 d, e, f cannot be easily recognized.

3. P 6, line 14: background with $\theta_c = 90^\circ$ (hopfions, Fig. 2d) under the confinement
> $\theta_c = 0^\circ$?

4. The Extended Data Fig. 2 a, Fig. 3 a, b need to be redrawn as they seem to have some errors.

Figure 2 a-c as well as Extended Data Fig. 1a, Fig. 2a, and Fig. 3a are the same diagram, but they can be confusing because Toron suddenly appears.

5. Line 4 of Fig. 3: θ_c at midplane increases from 0° to 90° as the applied voltage increases
> θ_c at midplane decreases from 90° to 0° as the applied voltage increases?

6. Line 7 of Fig. 4: Hopf soliton shown in (b-c)

> Hopf soliton shown in (a-b) ?

7. Figure 4c needs to be redrawn as it seems to have errors.
8. Line 11 of Fig. 4: The inset and the red line in (e) correspond to hopfions.

> What does this line mean?

9. Line 23 of Fig. 4: bars are 10 μm in (a-c)

> bars are 10 μm in (a and d)?

Main text:

1. P5, line 8-9: bend, twist, and splay deformations

> splay, twist, bend

2. P6, line 2-4 : Hopf solitons are found to have $n(r)$ with lower energy than their embedding backgrounds in parameter regions between those of metastable Hopf solitons and across the helical-uniform and helical-conical boundaries.

> What does this line mean? Where is the helical-uniform boundary?

Figures:

1. Not only do the structural stability diagrams in Figure 2 lack the definition of stable, meta stable, unstable, but they also do not have a clear explanation of the contour lines of background mid-plane cone angle θ_c .

For example, what does it mean to be in a meta stable state? If the Hopion is created in a E-field through inter-transformation, will it be maintained even if the E-field is turned off? The structural stability diagrams in Fig. 2 lack detail in their explanations.

2. Figure 2a, b does not have descriptions regarding e, d, g in the caption, so their relation with Fig. 2 d, e, f cannot be easily recognized.

3. P 6, line 14: background with $\theta_c = 90^\circ$ (hopfions, Fig. 2d) under the confinement

> $\theta_c = 0^\circ$?

4. The Extended Data Fig. 2 a, Fig. 3 a, b need to be redrawn as they seem to have some errors.

Figure 2 a-c as well as Extended Data Fig. 1a, Fig. 2a, and Fig. 3a are the same diagram, but they can be confusing because Toron suddenly appears.

5. Line 4 of Fig. 3: θ_c at midplane increases from 0° to 90° as the applied voltage increases

> θ_c at midplane decreases from 90° to 0° as the applied voltage increases?

6. Line 7 of Fig. 4 : Hopf soliton shown in (b-c)

> Hopf soliton shown in (a-b) ?

7. Figure 4c needs to be redrawn as it seems to have errors.

8. Line 23 of Fig. 4: bars are 10 μm in (a-c)

> bars are 10 μm in (a and d)?

Main text:

1. P5, line 8-9: bend, twist, and splay deformations

> splay, twist, bend

2. P6, line 2-4 : Hopf solitons are found to have $n(r)$ with lower energy than their embedding backgrounds in parameter regions between those of metastable Hopf solitons and across the helical-uniform and helical-conical boundaries.

> What does this line mean? Where is the helical-uniform boundary?

3. P8, line 2 We experimentally generated a heliknoton with laser tweezers

> This is not in the methods or in the references.

Reviewer #2 (Remarks to the Author):

This paper deal with the experimental realization and numerical calculation on the topology in cholesterics at various anchoring conditions. They provide appropriate range of parameters such elastic constant and electric field strength to study the topology tuning in their simulations and reported unique dynamics of hopf solitons. This paper is also written in a comprehensible manner. The authors' work adds an important contribution to soft matter physics and materials science by providing suitable design for fabricating and manipulating hopf solitons. I am sure this work will attract considerable interest in the community of in liquid crystal and relevant fields. There are several issues to be addressed before it is accepted in Nature Communications. 1. It is not clear what kind initial conditions were used for the simulations, especially for the confined bulk cholesterics (e.g. Fig. 1d). One may suppose they start from a random director field, and there is no BC applied (they claimed as it is). However, it looks like they have a homeotropic BC. Is it a just coincident outcome? 2. It is not very clear the relationship between the magnetic and cholesteric system. More discussions are needed. 3. Fig. 2a-c. Stable hopf solitons are surrounded either by metastable or unstable state depending on the elastic anisotropy or electric field strength. Does it mean the electricfield-driven transition can be either first-order or second-order like? Did they see it by tracking time evolution of the free energy, e.g. see if there is an energy barrier separating (meta)stable states? To my curiosity, what would be the reasons for the change of the transition order? 4. Figs. 3, S7. The transformation is interesting and impressive. Though they showed the simulated preimages, the director field is missing, which is surely needed to better understand the variation of the entanglement. Also as they showed, the electric-field response is

asymmetric in structure and time. The backward transformation is slowed down compared with forward transformation. For the transport of these particles, I suppose the time-asymmetry serves as the main driving force. It is true? Can more discussions be added? Furthermore, it would be nice if the authors can suggest if an asymmetric electric field with different time periods for positive and negative pulses can be used to control the transport speed or even add a delay between the positive and negative pulses? 5. Fig. 3. It is helpful to show the simulation time of the so-called 100% of time. 6. It is helpful to partially explain the effect of K11 on the topology, though they may address the issue in future studies. 7. The explanation of “theta_c in the middle” needs more words to explain. Through the paper, though I can understand, the descriptions about theta_c are not very clear

Reviewer #3 (Remarks to the Author):

This article is a experimental and computational investigation into the stability of Hopf solitons in chiral liquid crystalline (and ferromagnetic) systems, under changes in applied field, boundary conditions and far-field behaviour, as well as a study of the transition pathways between these states, which induces an interesting squirming motion.

The work appears competently performed and I have no problems with the technical aspects.

Hopf solitons have broad relevance beyond liquid crystals into other areas of physics, mathematics and materials science, and their study is appropriate topic for a journal such as Nature Communications.

A large amount work has been done on a variety of similar solitons in liquid crystals by this and other groups, and much of the phenomena reported are not new. Having said that, transitions between these kinds of structures is an underexplored area, and the novel field-induced squirming motion is very interesting. In terms of quality and novelty I would judge this work as good enough for Nature Communications.

As such, I recommend publication, subject to the following changes.

1. Structures related to 'heliknotons' albeit with defects, were found numerically [1] and this reference should be cited.

2. One of the authors also wrote an early paper generating Hopf solitons [2], which should also be cited.

3. Page 5 lines 19-20, the authors write that for $E=0$, heliknotons have energy density higher than the helical background when $\text{Sqrt}(K_{33}/K_{22}) \leq 1.2$, implying that it is lower otherwise. I think they mean that the heliknotons are unstable rather than metastable in this regime, the energy density of the helical background at $E=0$ is a constant proportional to $-(1/p_0)^2$ in this model, which is the minimum possible value, so the statement given by the authors is not correct as written (indeed there is no stable region for $E=0$ in Fig 2a).

4. The authors talk about stability through confinement when the background texture is uniform, as well as a related phenomena of 'the soliton filling the available computational domain'. This point is made on page 6 (line 5) and page 10 (also line 5).

In a uniform background, this phenomenon follows from a simple Derrick theorem type scaling argument. The authors should include this argument, and verify that their results are consistent with it. Such an analysis will find that in a uniform background, a soliton with an $n \cdot \text{curl } n$ term in the energy will either shrink to nothing or otherwise expand to fill all available space (provided it can expand in all three directions). This explains the 'soliton filling the available computational domain' phenomenon.

5. The authors should elaborate on the direction of motion of the soliton, forward versus backwards. The experimental and computational results show a consistent direction of travel.

Looking at, Figure 2 e, middle panel, a $n \rightarrow -n$ transformation (which is a symmetry of a director field) followed by a 180 degree rotation about the z axis leaves the heliknoton invariant.

The direction of motion must therefore be induced by the boundary conditions on the top and bottom of the cell (Fig 3a), which distinguish the white and black lines in Fig 3f.

i.e. a pure heliknoton in an unconfined system has an unoriented long axis, it is only the homeotropic anchoring of the cell that allows one to define an oriented long axis.

6. I find Fig 3f, which shows the pathway for the transition, unsatisfying. The middle panel is unclear to me, and seems to be where all the interesting parts of the transition take place. The authors might like to find an alternative mechanism to better visualise the pathway.

[1] Machon, T., & Alexander, G. P. (2014). Knotted defects in nematic liquid crystals. *Physical review letters*, 113(2), 027801.

[2] Chen, B. G. G., Ackerman, P. J., Alexander, G. P., Kamien, R. D., & Smalyukh, I. I. (2013). Generating the Hopf fibration experimentally in nematic liquid crystals. *Physical review letters*, 110(23), 237801.

Geometric transformation and three-dimensional hopping of Hopf solitons

Jung-Shen B. Tai, Jin-Sheng Wu, Ivan I. Smalyukh

Summary of Changes and Response to Reviewers' Requests/Remarks

To editor:

Thank you for handling our manuscript. We are happy to see all three reviewers evaluated our manuscript positively and suggested our work for publication in *Nature Communications* after the comments by the reviewers are addressed. We have accounted for these comments fully in this report. For clarity, we've highlighted changes made in the revised manuscript in yellow. In the revision, we reformatted our manuscript following *Nature communications'* formatting instructions. In particular, an abstract is provided and the introduction is modestly modified according to the new guidelines. Additionally, in response to reviewers' comments, two additional main text figures (Figs. 3 and 5 in the revised manuscript) were added to the revised manuscript for better presentation and visualization. Fig. 4 in the original manuscript was split into two figures (Figs. 6 and 7 in the revised manuscript) to aid in clarity. The Supplementary Information was modified accordingly without adding new data. We have highlighted these parts of the manuscript to make sure these revisions are easily accessible to the editor and the reviewers.

Report of Reviewer 1:

They have shown, for the first time, that Hopf solitons can exist as spatially localized structures without external fields or confinement in the helical background of chiral LCs within the proper range of elastic anisotropy parameters, as well as that they can be stable or metastable in the conical backgrounds when an external electric field is applied.

However, it is questionable whether making their previously made Hopf soliton meta stable is noble. Nevertheless, a pathway for inter-transformation between hopfions and heliknotons as well as for 3D hopping-like dynamics have been identified in detail and the simulated data is considerable. Accordingly, I think that an impact can be made in expanding the applicability of Hopf solitons and in engineering related to materials.

Author:

We thank Reviewer 1 for the positive and constructive review of our manuscript, stating that “*a pathway for inter-transformation between hopfions and heliknotons as well as for 3D hopping-like dynamics have been identified in detail and the simulated data is considerable*” and “*an impact can be made in expanding the applicability of Hopf solitons and in engineering related to materials.*” We also thank Reviewer 1's helpful remarks that improve the manuscript, which we account for in detail point-by-point below.

Reviewer 1:

I recommend the following needs revision.

Figures:

1. Not only do the structural stability diagrams in Figure 2 lack the definition of stable, meta stable, unstable, but they also do not have a clear explanation of the contour lines of background mid-plane cone angle θ_c .

For example, what does it mean to be in a meta stable state? If the Hopion is created in a E-field through inter-transformation, will it be maintained even if the E-field is turned off? The structural stability diagrams in Fig. 2 lack detail in their explanations.

Author:

We thank Reviewer 1 for this remark. In constructing the structural stability diagrams, we start from topologically nontrivial heliknoton or hopfion states based on experimental reconstruction (see our Science 2019 article) as the initial states and determine if the final energy-minimizing structure has the same topology. If topology is preserved, based on comparing the energy of the solitonic structure relative to the embedding background and other structures, we classify Hopf solitons with energy lower or higher than the background state as stable or metastable, respectively (corresponding to global or local energy minima, respectively). The field structure is classified as unstable if the final structure does not share the same topology as the initial heliknoton or hopfion, e.g. forming topological trivial states or singular defects. The applied \mathbf{E} field is not changed throughout the simulations in constructing the stability diagrams in Fig. 2. The background mid-plane cone angle θ_c refers to the cone angle between the vectorized background director field and the \hat{z} direction (corresponding to the direction of the applied \mathbf{E} field or the bulk helical axis χ_0) in the mid-plane of the computational volume. We have added explanations regarding the classification of unstable, metastable, and stable states for Fig. 2, a-c as well as properly introduced the cone angle θ_c in the revised manuscript. We thank Reviewer 1 once again for these remarks that help improve the clarity of our manuscript.

Reviewer 1:

2. Figure 2a, b does not have descriptions regarding e, d, g in the caption, so their relation with Fig. 2 d, e, f cannot be easily recognized.

Author:

We thank Reviewer 1 for this suggestion. We have added the corresponding descriptions in the caption of Figure 2 as well as in Supplementary Figs. 1-4 where we used similar labeling.

Reviewer 1:

3. P 6, line 14: background with $\theta_c = 90^\circ$ (hopfions, Fig. 2d) under the confinement
> $\theta_c = 0^\circ$?

Author:

We thank Reviewer for pointing this out. We have made the corresponding correction in the revised manuscript.

Reviewer 1:

4. The Extended Data Fig. 2 a, Fig. 3 a, b need to be redrawn as they seem to have some errors.

Figure 2 a-c as well as Extended Data Fig. 1a, Fig. 2a, and Fig. 3a are the same diagram, but they can be confusing because Toron suddenly appears.

Author:

We thank the reviewer for pointing out the errors in the figures. We have redrawn and replaced the original figures in the revised manuscript. We have also added describing text on top of each diagram in Fig. 2, a-c where stability diagrams of various materials and confinement conditions are presented to ensure this is clear to the readers. We also thank Reviewer 1 for the suggestion in improving the clarity of Fig. 2, a-c and Supplementary Fig. 1a, 2a and 3a. In the revised manuscript, to assure clarity, we use “unstable (toron)” and “unstable (trivial)” to label toron and trivial states in the supplementary figures where these two unstable regions are distinguished. We thank Reviewer 1 again for these helpful suggestions that improve the accessibility of our work.

Reviewer 1:

5. Line 4 of Fig. 3: θ_c at midplane increases from 0° to 90° as the applied voltage increases
> θ_c at midplane decreases from 90° to 0° as the applied voltage increases?

Author:

We thank Reviewer 1 for pointing this out. We have made the corresponding correction in the revised manuscript.

Reviewer 1:

6. Line 7 of Fig. 4: Hopf soliton shown in (b-c)
> Hopf soliton shown in (a-b) ?

Author:

We thank Reviewer 1 for pointing this out. We have made the corresponding correction in the revised manuscript.

Reviewer 1:

7. Figure 4c needs to be redrawn as it seems to have errors.

Author:

We thank Reviewer 1 for pointing this out. We have made the corresponding correction to the figure in the revised manuscript.

Reviewer 1:

8. Line 11 of Fig. 4: The inset and the red line in (e) correspond to hopfions.
> What does this line mean?

Author:

We thank Reviewer 1 for this question. In Fig. 4e in the original manuscript, the blue and orange data points in (e) show the relation between $z_f - z_i$ and z_i for heliknotons with $d = 3p_0$ and $d = 4p_0$, and the inset and the red line in (e) correspond to the relation between $z_f - z_i$ and z_i for hopfions in a uniform background. For better clarity, in the revised manuscript, we have split this figure part into Fig. 6e,f where hopfions and helikntons are presented separately. We also rewrote the corresponding caption. We thank Reviewer 1's question that direct the changes that improve the clarity of our manuscript.

Reviewer 1:

9. Line 23 of Fig. 4: bars are 10 μm in (a-c)

> bars are 10 μm in (a and d)?

Author:

We thank Reviewer 1 for pointing this out. We have made the corresponding correction to the caption in the revised manuscript.

Reviewer 1:

Main text:

1. P5, line 8-9: bend, twist, and splay deformations

> splay, twist, bend

Author:

We thank Reviewer 1 for pointing this out. We have made the corresponding correction in the revised manuscript.

Reviewer 1:

2. P6, line 2-4 : Hopf solitons are found to have $n(r)$ with lower energy than their embedding backgrounds in parameter regions between those of metastable Hopf solitons and across the helical-uniform and helical-conical boundaries.

> What does this line mean? Where is the helical-uniform boundary?

Author:

We thank the reviewer for this question. The sentence describes where Hopf solitons with lower energy than their embedding backgrounds are found on the structural stability diagram – between patches of parameter regions of metastable Hopf solitons and across the helical-uniform and helical-conical boundaries for the background field. We have modified the sentence in the revised manuscript accordingly to assure this is clear to the readers.

Reviewer 1:

3. P8, line 2 We experimentally generated a heliknoton with laser tweezers

> This is not in the methods or in the references.

Author:

We thank Reviewer 1 for this remark. We draw the reviewer's attention to the "Laser generation and imaging of Hopf solitons" part in the methods section, now on page 16 of the revised main-text manuscript, where the method of generating Hopf solitons using laser tweezers and relevant references are detailed. We also assure the readers are referred to the Methods section in the revised manuscript. We thank the reviewer's comment that makes our work more accessible to the readers.

Report of Reviewer 2:

This paper deal with the experimental realization and numerical calculation on the topology in cholesterics at various anchoring conditions. They provide appropriate range of parameters such elastic constant and electric field strength to study the topology tuning in their simulations and reported unique dynamics of hopf solitons. This paper is also written in a comprehensible manner.

The authors' work adds an important contribution to soft matter physics and materials science by providing suitable design for fabricating and manipulating hopf solitons. I am sure this work will attract considerable interest in the community of in liquid crystal and relevant fields.

Author:

We thank Reviewer 2's positive evaluation of our work, stating that our work "adds an important contribution to soft matter physics and materials science by providing suitable design for fabricating and manipulating hopf solitons" and "will attract considerable interest in the community of in liquid crystal and relevant fields." We also thank the remarks raised by Reviewer 2 which we address in detail point-by-point in the following.

Reviewer 2:

There are several issues to be addressed before it is accepted in Nature Communications.

1. It is not clear what kind initial conditions were used for the simulations, especially for the confined bulk cholesterics (e.g. Fig. 1d). One may suppose they start from a random director field, and there is no BC applied (they claimed as it is). However, it looks like they have a homeotropic BC. Is it a just coincident outcome?

Author:

We thank Reviewer 2 for this comment. We used Hopf soliton structures, first experimentally reconstructed and numerically relaxed as described previously (see our 2019 Science article, where we also utilized such methodology) as initial conditions in the simulations to arrive at the structural stability diagrams shown in Figs. 2a-c and Supplementary Figs. 1a, 2a, 3a, and 4a. We have provided additional explanations in the main text to assure this is clear to the readers. In this work, we investigated structural diagrams with different confinement (bulk vs. confined) and boundary (perpendicular vs. parallel) conditions. For example, Fig. 2a was simulated in a bulk with no boundary conditions while Fig. 2b was simulated with confinement and perpendicular boundary conditions at the confining surfaces. To aid in clarity of the specific confinement and boundary conditions used for each diagram, we added extra labeling texts on top of each structural diagram in Fig. 2 where stability diagrams of various materials and confinement conditions are shown together. We thank Reviewer 2 again for this remark that helps improve clarity of our manuscript.

Reviewer 2:

2. It is not very clear the relationship between the magnetic and cholesteric system. More discussions are needed.

Author:

We thank Reviewer 2 for this remark. We study if structural stability and geometric transformations of Hopf solitons in liquid crystals can be extended to other condensed materials by performing similar stability analysis of Hopf solitons in chiral magnets. Chiral liquid crystals and chiral magnets share similarities in their respective continuum energetic functionals – Frank-Oseen free energy and

micromagnetic Hamiltonian. In fact, the micromagnetic Hamiltonian of chiral magnets can be obtained from Frank-Oseen free energy after adopting the so-called one-constant approximation where $K_{11} = K_{22} = K_{33}$. With this approximation, the average elastic constant K and the twisting coefficient $2\pi K/p_0$ in liquid crystals can be directly related to the exchange and Dzyaloshinskii-Moriya interaction constants, respectively. Therefore, similarities in energetics and structures can be anticipated in these distinct systems. Indeed, beyond similarities, there exist properties unique to either material system such as magnetocrystalline anisotropies, Zeeman energy and nonlocal dipole-dipole interaction in magnets (and we have indeed considered such effects in our past studies like *Phys. Rev. Lett.* **125**, 057201 (2020) & *Phys. Rev. Lett.* **121**, 187201 (2018)), though relevance of these effects depends on specific geometries and material systems and is outside of the scope of this present work that focuses on liquid crystals and draws only the general connection to magnetic solids. We have supplemented corresponding sentences in the main text and added discussions of these aspects in the Methods section in the revised manuscript to assure this is accessible to the readers. We thank again Reviewer 2's thoughtful comment that helps improve the accessibility of our manuscript.

Reviewer 2:

3. Fig. 2a-c. Stable hopf solitons are surrounded either by metastable or unstable state depending on the elastic anisotropy or electric field strength. Does it mean the electric- field-driven transition can be either first-order or second-order like? Did they see it by tracking time evolution of the free energy, e.g. see if there is an energy barrier separating (meta)stable states? To my curiosity, what would be the reasons for the change of the transition order?

Author:

We thank Reviewer 2 for this question. In our simulations, the transitions between metastable/stable regions are continuous and second-order-like both in energy and field configurations, while the transitions between metastable/unstable and stable/unstable are more abrupt and first-order-like. The difference lies in the topology of the field configurations, where the topology of the field configurations in the metastable and stable regions is nontrivial and classified by the same Hopf index, while field configurations in the unstable region is either topologically trivial or contain singular defects. This implies a transition from (meta)stable region to unstable region cannot be continuous and has to go through generation and/or annihilation of singular defects, which contribute to the emergence of abrupt first-order-like transition. However, how an order-parameter reflecting the topological state of matter and the corresponding transition order can be properly defined and systematically analyzed await future studies and are beyond the scope of this work, though indeed of interest for future studies. We also note our simulations investigate the structural stability by energy minimization; for a metastable Hopf soliton with energy higher than the embedding background, there exists energetic barriers preventing the relaxation of the Hopf soliton into the topologically trivial background, and the origin of the barrier is topological. For the states undergoing transition, such as the geometric transformation induced by changing the electric field, each geometric manifestation of the Hopf soliton relaxes into one another upon a change in the energetic landscape upon the change in the electric field. We thank again Reviewer 2's thoughtful remark.

Reviewer 2:

4. Figs. 3, S7. The transformation is interesting and impressive. Though they showed the simulated preimages, the director field is missing, which is surely needed to better understand the variation of the entanglement. Also as they showed, the electric-field response is asymmetric in structure and time. The backward transformation is slowed down compared with forward transformation. For the transport of

these particles, I suppose the time-asymmetry serves as the main driving force. It is true? Can more discussions be added? Furthermore, it would be nice if the authors can suggest if an asymmetric electric field with different time periods for positive and negative pulses can be used to control the transport speed or even add a delay between the positive and negative pulses?

Author:

We thank Reviewer 2 for this remark. To respond to the reviewer's first part of the remark, due to the constraint of space, we only show the preimages of the geometric transformation process in the original main-text Fig. 3 (Fig. 4 in revised manuscript). However, we do present the director fields in different cross-sections during the transformation in Supplementary Fig. 5. We have added another reference to Supplementary Fig. 5 in the main text to make sure this is accessible to the readers. Additionally, related to another comment by Reviewer 3, we provide additional detailed visualizations of preimages during the transformation as Fig. 5 in the revised manuscript. We trust the transformation process is accessible to the readers after these modifications in the revised manuscript.

Regarding the second part of the reviewer's remark, indeed, asymmetry in the evolution of field configurations between the forward and backward transformations is necessary for the translation of the particle, as a perfectly time-symmetric transformation will only result in pulsing of the soliton. In 2D skyrmions, experiments and numerical modeling [Ackerman et al. (2017)] and a coarse-grained theory [Long et al. (2021)] showed 2D skyrmions move in opposite directions but with different magnitudes, upon application and removal of an electric field, thus generating a rectified motion. In 3D Hopf solitons, more drastic asymmetry in the field configurations between backward and forward transformations was found in both experiments and simulations, resulting in the directions of motion to be consistent in either direction of transformation. We have expanded the relevant discussions in the revised manuscript. However, a mechanistic study of the transport of the Hopf soliton, such as one involving a 3D coarse-grained theory, requires future studies and are beyond the scope of this paper. Regarding the pattern of electric field modulation for soliton motions, in this work, we show that 3D hopping-like motion of Hopf solitons can be achieved with the simplest ON/OFF modulation of the applied electric field. A more complex voltage modulation profile, such as the ones including different time of the ON/OFF periods, delays, or arbitrary modulation profile, can indeed be used to further optimize the speed or even the direction of Hopf soliton transport. We have added relevant discussions in the revised manuscript. We thank Reviewer 2 again for this remark.

Reviewer 2:

5. Fig. 3. It is helpful to show the simulation time of the so-called 100% of time.

Author:

We thank Reviewer 2 for this suggestion. The simulated time can be mapped well to the physical time in experiments using the iteration time step and rotational viscosity of the LC material. Therefore, the total simulation time (100%) agrees with the total transformation time in the experiments. We have added this clarification in the caption of Fig. 4 in the revised manuscript. We thank Reviewer 2 again for the thoughtful remark that helps improve the clarity of our manuscript.

Reviewer 2:

6. It is helpful to partially explain the effect of K_{11} on the topology, though they may address the issue in future studies.

Author:

We thank Reviewer 2 for this remark. In this work, we investigate the effect of adjusting bend elastic constant K_{33} on the stability of Hopf solitons in chiral LCs, due to the accessibility of modulating K_{33} using a mixture of rod-like and bent-core LC molecules, as well as the fact that K_{33} typically has the largest magnitude among elastic constants in LCs made of rod-like molecules, thus the effect being most prominent. Indeed, the other degree of freedom, which is the magnitude of splay elastic constant K_{11} , can also be engineered and change the stability of solitons independently or in coordination with the engineering of K_{33} , which requires future studies. We have added this discussion to the main text. We thank Reviewer 2 again for this thoughtful comment.

Reviewer 2:

7. The explanation of “theta_c in the middle” needs more words to explain. Through the paper, though I can understand, the descriptions about theta_c are not very clear.

Author:

We thank Reviewer 2 for this comment. We have added extra explanations to the revised manuscript of the definition of cone angle θ_c for bulk simulations and simulations with confinement. We trust it is clear to the readers in the revised form. We thank Reviewer 2 for bringing this up that help improve the clarity of our manuscript.

Report of Reviewer 3:

This article is a experimental and computational investigation into the stability of Hopf solitons in chiral liquid crystalline (and ferromagnetic) systems, under changes in applied field, boundary conditions and far-field behaviour, as well as a study of the transition pathways between these states, which induces an interesting squirming motion.

The work appears competently performed and I have no problems with the technical aspects.

Hopf solitons have broad relevance beyond liquid crystals into other areas of physics, mathematics and materials science, and their study is appropriate topic for a journal such as Nature Communications.

A large amount work has been done on a variety of similar solitons in liquid crystals by this and other groups, and much of the phenomena reported are not new. Having said that, transitions between these kinds of structures is an underexplored area, and the novel field-induced squirming motion is very interesting. In terms of quality and novelty I would judge this work as good enough for Nature Communications.

As such, I recommend publication, subject to the following changes.

Author:

We thank Reviewer 3 for the positive evaluations of our work, stating that our work is “*completely performed*”, and “*the novel field-induced squirming motion is very interesting,*” as well as stating that “*Hopf solitons have broad relevance beyond liquid crystals into other areas of physics, mathematics and materials science, and their study is appropriate topic for a journal such as Nature Communications*” and recommending for publication in terms of “*quality and novelty*”. We also thank the remarks raised by Reviewer 3 which we address in detail point-by-point in the following.

Reviewer 3:

- 1. Structures related to 'heliknotons' albeit with defects, were found numerically [1] and this reference should be cited.*
- 2. One of the authors also wrote an early paper generating Hopf solitons [2], which should also be cited.*

Author:

We thank Reviewer 3 for these suggestions. We agree these are relevant references and have added them to the revised manuscript.

Reviewer 3:

- 3. Page 5 lines 19-20, the authors write that for $E=0$, heliknotons have energy density higher than the helical background when $\sqrt{K_{33}/K_{22}} \leq 1.2$, implying that it is lower otherwise. I think they mean that the heliknotons are unstable rather than metastable in this regime, the energy density of the helical background at $E=0$ is a constant proportional to $-(1/p_0)^2$ in this model, which is the minimum possible value, so the statement given by the authors is not correct as written (indeed there is no stable region for $E=0$ in Fig 2a).*

Author:

We thank Reviewer 3 for this comment. We do mean the solitons are either metastable or unstable at no electric field in an unconfined bulk. We have modified the wording in the revised manuscript as the following: “At $\tilde{E} = 0$, Hopf solitons are metastable in the helical background (heliknotons) with higher

energy density than the background when $\sqrt{K_{33}/K_{22}} \leq 1.2$ and unstable otherwise (Fig. 2a,e).” We trust this is clear to the readers now. We thank Reviewer 3 again for this comment that help improve the clarity of our manuscript.

Reviewer 3:

4. The authors talk about stability through confinement when the background texture is uniform, as well as a related phenomena of 'the soliton filling the available computational domain'. This point is made on page 6 (line 5) and page 10 (also line 5).

In a uniform background, this phenomenon follows from a simple Derrick theorem type scaling argument. The authors should include this argument, and verify that their results are consistent with it. Such an analysis will find that in a uniform background, a soliton with an n dot curl n term in the energy will either shrink to nothing or otherwise expand to fill all available space (provided it can expand in all three directions). This explains the 'soliton filling the available computational domain' phenomenon.

Author:

We thank Reviewer 3 for this remark. Derrick’ theorem provides an elegant method to examine the stability of localized soliton solutions by a simple scaling argument of the underlying energy functional. It has been shown that in the simplest linear theories, localized solutions cannot be stable in higher dimensions. 3D Hopf solitons in this work are studied in a uniform or a modulated background (helical or conical background). In the former case, the relevant energy terms include the Frank-Oseen elastic distortion, chirality, and dielectric coupling. We note that an applied electric field and the corresponding dielectric coupling is necessary to align the director field into a uniform background, since the ground state of a chiral liquid crystal is a helical background at no field. The three energy terms and their scaling as λ^{-1} , λ^{-2} , and λ^{-3} in 3D (Here λ is the scaling factor), respectively, combine to prevent a simple prediction of stability by a scaling argument. In the other case where solitons are stabilized in a modulated background with an intrinsic length scale, the scaling argument of Derrick’s theorem does not straightforwardly apply, as a scaling operation changes the length scale of the background modulation and leads to divergence in energy. Furthermore, in the presence of confinement (as referred to by Reviewer 3 on page 10 in the original manuscript), the energy functional must include surface terms and becomes even more complex, which further prevents the application of Derrick’s theorem and effectively evades it. Therefore, due to the complications mentioned above, we choose not to interpret the observed phenomena as the outcome of a simple prediction of Derrick’s theorem, but simply as the result of energy minimization. We also believe that the analysis in relation to the Derrick’s theorem predictions can be of interest to future detailed studies.

The reason behind solitons having energy lower than the background and their expansion can be understood as the result of limitation on the degrees of freedom in the background states studied in the structural stability diagram compared to those of the solitonic structures. The background states are limited to two degrees of freedom – cone angle and helical pitch, upon changes in material elasticity and applied electric field. On the other hand, the solitonic structures are not constrained in structural degrees of freedom and simply evolves as a result of energy minimization. In the intermediate parameter regions where the background states are not at the extremes of the cone angle at low or high electric fields, solitons can readily take structures with lower energy density that the background state and expand to lower the overall free energy. We have added relevant discussions in the revised manuscript and trust this is clear to the readers now. We thank Reviewer 3 again for this thoughtful remark.

Reviewer 3:

5. The authors should elaborate on the direction of motion of the soliton, forward versus backwards. The experimental and computational results show a consistent direction of travel.

Looking at, Figure 2 e, middle panel, a $n \rightarrow -n$ transformation (which is a symmetry of a director field) followed by a 180 degree rotation about the z axis leaves the heliknoton invariant.

The direction of motion must therefore be induced by the boundary conditions on the top and bottom of the cell (Fig 3a), which distinguish the white and black lines in Fig 3f.

i.e. a pure heliknoton in an unconfined system has an unoriented long axis, it is only the homeotropic anchoring of the cell that allows one to define an oriented long axis.

Author:

We thank Reviewer 3 for this suggestion. Indeed, in a heliknoton embedded in a helical background, the two polar preimages are not distinguished due to the nonpolar $\mathbf{n} \equiv -\mathbf{n}$ symmetry. Therefore, the heliknoton has a two-fold rotational symmetry and the long axis is not oriented. It is only when the background state transitions into the uniform state that the symmetry is broken and the two preimages are distinguished – the one associated with the far-field background and, when the liquid crystal director is vectorized, and the one with the opposite vectorized orientation to the far-field background. When an aligning electric field is applied to a heliknoton embedded in an infinite bulk, the emergence of either polar preimage as the one associated with the far-field background should be equally probable. However, in a confined liquid crystal with perpendicular boundary conditions, the far-field is preselected by the boundary condition, leading to the consistent orientation of the heliknoton long axis and direction of translation. We have elaborated relevant discussions in the revised manuscript. We thank Reviewer 3 again for this suggestion that improve the clarity of our work.

Reviewer 3:

6. I find Fig 3f, which shows the pathway for the transition, unsatisfying. The middle panel is unclear to me, and seems to be where all the interesting parts of the transition take place. The authors might like to find an alternative mechanism to better visualise the pathway.

Author:

We thank Reviewer 3 for this suggestion. In order to make the transformation process as evident as possible, we supplement the original Fig. 3f (Fig. 4f in the revised manuscript) with Supplementary Video 5, where the evolution of the north- and south-pole preimages are visualized throughout the transformation. To make the transformation process even more accessible, we additionally include Fig. 5 in the revised manuscript, where the transformation is presented in snapshots of more preimages in addition to the polar preimages, with focus on the middle part where the most complex processes take place. We believe this allows the visualization of the transformation much more accessible to the readers. We thank Reviewer 3 again for this comment that help improve the accessibility of our work.

References added:

[17] Chen, B. G. G., Ackerman, P. J., Alexander, G. P., Kamien, R. D., & Smalyukh, I. I. Generating the Hopf fibration experimentally in nematic liquid crystals. *Phys. Rev. Lett.*, **110**, 237801.

[18] Machon, T., & Alexander, G. P. Knotted defects in nematic liquid crystals. *Phys. Rev. Lett.*, **113**, 027801 (2014).

[35] Long, C. & Selinger, J. V. Coarse-grained theory for motion of solitons and skyrmions in liquid crystals. *Soft Matter* **17**, 10437–10446 (2021).

[42] Wu, J.-S. & Smalyukh, I. I. Hopfions, heliknotons, skyrmions, torons and both abelian and nonabelian vortices in chiral liquid crystals. *Liq. Cryst. Rev.* 1–35 (2022).

[43] Leonov, A. O. Surface anchoring as a control parameter for shaping skyrmion or toron properties in thin layers of chiral nematic liquid crystals and noncentrosymmetric magnets. *Phys. Rev. E* **104**, 044701 (2021).

We believe that the above-described revisions address all remarks and account for all suggestions that were raised for the original manuscript. These manuscript changes introduced during this revision are highlighted in yellow. We feel that the revisions improve the paper significantly. We would like to thank all reviewers for the thoughtful and valuable comments that helped direct these changes. We trust that this revised version is now well suited for publication in *Nature Communications*.

Sincerely,

on behalf of authors,

Prof. Ivan I. Smalyukh, Department of Physics, 390 UCB

Senior Investigator, Liquid Crystal Materials Research Center (LCMRC)

University of Colorado at Boulder

Phone: 303-492-7277 (office); Email: Ivan.Smalyukh@colorado.EDU

<http://www.colorado.edu/physics/SmalyukhLab/index.html>

REVIEWERS' COMMENTS

Reviewer #1 (Remarks to the Author):

The manuscript has improved a lot through the previous review. Therefore, I am positive about the publication of this paper and have no further opinions.

Reviewer #2 (Remarks to the Author):

The authors have responded thoroughly to the comments of all the referees and the manuscript is much improved. Now, I believe the paper is now suitable for acceptance.

Reviewer #3 (Remarks to the Author):

The revised manuscript is greatly improved, I appreciate the added comments and new Figures, and I am happy to recommend it for publication.

The authors did not include my suggestion of a scaling argument (although my comments were perhaps somewhat cryptic), but I am satisfied with the response given.

Geometric transformation and three-dimensional hopping of Hopf solitons

Jung-Shen B. Tai, Jin-Sheng Wu, Ivan I. Smalyukh

Summary of Changes and Response to Reviewers' Requests/Remarks

Report of Reviewer 1:

The manuscript has improved a lot through the previous review. Therefore, I am positive about the publication of this paper and have no further opinions.

Report of Reviewer 2:

The authors have responded thoroughly to the comments of all the referees and the manuscript is much improved. Now, I believe the paper is now suitable for acceptance.

Report of Reviewer 3:

The revised manuscript is greatly improved, I appreciate the added comments and new Figures, and I am happy to recommend it for publication.

The authors did not include my suggestion of a scaling argument (although my comments were perhaps somewhat cryptic), but I am satisfied with the response given.

Author:

We are happy to see all three reviewers are satisfied with our response to their comments and the revision of the manuscript. We also appreciate all reviewers' recommendation for publication. The revised manuscript was formatted to comply with editorial requirements and we trust that this current version is well suited for publication in *Nature Communications*.

Sincerely,

on behalf of authors,

Prof. Ivan I. Smalyukh, Department of Physics, 390 UCB

Senior Investigator, Liquid Crystal Materials Research Center (LCMRC)

University of Colorado at Boulder

Phone: 303-492-7277 (office); Email: Ivan.Smalyukh@colorado.EDU

<http://www.colorado.edu/physics/SmalyukhLab/index.html>